# The relative resistance of children to sepsis mortality: from pathways to drug candidates

Rose B Joachim[1,†], Gabriel M Altschuler[2,†], John N Hutchinson[3] , Hector R Wong[4], Winston A Hide[2,3,‡,*] & Lester Kobzik[1,5,‡,**]

## Abstract

Attempts to develop drugs that address sepsis based on leads developed in animal models have failed. We sought to identify leads based on human data by exploiting a natural experiment: the relative resistance of children to mortality from severe infections and sepsis. Using public datasets, we identified key differences in pathway activity (Pathprint) in blood transcriptome profiles of septic adults and children. To find drugs that could promote beneficial (child) pathways or inhibit harmful (adult) ones, we built an *in silico* pathway drug network (PDN) using expression correlation between drug, disease, and pathway gene signatures across 58,475 microarrays. Specific pathway clusters from children or adults were assessed for correlation with drug-based signatures. Validation by literature curation and by direct testing in an endotoxemia model of murine sepsis of the most correlated drug candidates demonstrated that the Pathprint-PDN methodology is more effective at generating positive drug leads than gene-level methods (e.g., CMap). Pathway-centric Pathprint-PDN is a powerful new way to identify drug candidates for intervention against sepsis and provides direct insight into pathways that may determine survival.

**Keywords** connectivity map; drug discovery; pathways; sepsis
**Subject Categories** Network Biology; Pharmacology & Drug Discovery; Systems Medicine
**Mol Syst Biol. (2018) 14: e7998**

See also: **S Timmermans & C Libert** (May 2018)

## Introduction

Sepsis is a major cause of global morbidity and mortality for which there remains no targeted therapy (Opal, 2014; Seymour & Rosengart, 2015; Weiss *et al*, 2015). Central to sepsis pathophysiology is a dysregulated host inflammatory response (Aziz *et al*, 2013; Wiersinga *et al*, 2014; Singer *et al*, 2016), suggesting that host-directed immunomodulators could be of therapeutic benefit (Delano & Ward, 2016). There is little agreement or certainty about which particular cells or molecules are critical to defining sepsis outcomes (Marshall, 2014). As a result, transcriptome analyses and systems biology approaches have been eagerly embraced as better ways to identify drug targets for sepsis (Maslove & Wong, 2014; Sweeney *et al*, 2015; Wong *et al*, 2015; Davenport *et al*, 2016). Systematic computational analysis represents an exciting class of approaches for prediction and discovery of novel targets and therapeutic indications (Dubus *et al*, 2009; Dudley *et al*, 2011; Hurle *et al*, 2013) reflecting their ability to provide virtual access to large numbers of compounds and data relating to the target disease (Kim, 2015).

However, the hope that "omics-based approaches" might guide the selection of promising therapeutics to target sepsis has not yet been realized. This is despite the fact that tools like the Connectivity Map (CMap) and Library of Network-Based Cellular Signatures (LINCS; Lamb *et al*, 2006; Prathipati & Mizuguchi, 2015), which use gene expression signatures to identify drug candidates, have been available for over a decade. Obstacles to progress in developing interventions for sepsis include discordant results across human studies focused on gene-level changes (Sweeney & Khatri, 2016), as well as the strongly debated limitations of animal models of sepsis for these types of analyses (Seok *et al*, 2013; Osuchowski *et al*, 2014). Here, we address these problems by using available data on human transcriptomes together with a powerful new approach that combines pathway-level analysis of human transcriptome samples with subsequent *in vivo* verification of findings in an animal model. We postulate that this "human-data-first" approach can improve results compared to prior efforts that began with findings in animal models. Our pathway-level analysis exploits a natural phenomenon in humans to directly compare two groups with widely disparate rates of survival from sepsis—children and adults. Using novel pathway-centered bioinformatic tools to optimize data analysis across

1 Department of Environmental Health, Harvard T.H. Chan School of Public Health, Boston, MA, USA
2 Department of Neuroscience, Sheffield Institute for Translational Neurosciences, University of Sheffield, Sheffield, UK
3 Department of Biostatistics, Harvard T.H. Chan School of Public Health, Boston, MA, USA
4 Division of Critical Care Medicine, Cincinnati Children's Hospital Medical Center, Cincinnati, OH, USA
5 Department of Pathology, Brigham & Women's Hospital, Boston, MA, USA
 *Corresponding author. Tel: +44 11 42222233; E-mail: winhide@sheffield.ac.uk
 **Corresponding author. Tel: +1 617 4322247; E-mail: lkobzik@hsph.harvard.edu
 †These authors contributed equally to this work
 ‡These authors contributed equally to this work

multiple platforms, we were able to identify key differences in the responses of both age groups to sepsis as well as identify potential therapeutics.

The comparison of data from septic children and adults arose from a striking finding, which at first glance seems unrelated to the problem of sepsis. Despite similar rates of infection during the 1918 influenza pandemic, children aged 5–14 experienced a remarkably lower rate of mortality compared to adults, dubbed the "honeymoon period" (Ahmed *et al*, 2007). Puberty (~ age 14 in the early 1900s) marked the age range in which mortality increased, suggesting that sex hormones could influence changes in fatality rates. Importantly, the "honeymoon period" is not limited to 1918 influenza-related resistance to mortality. Historical mortality rates are much lower in children after various high-fatality challenges, spanning from bubonic plague to measles. Contemporary data for trauma, the recent Ebola outbreaks, and other severe infections (Table 1) confirm the resistance. In particular, these data include lower case fatality rates for children with sepsis, both when linked to specific pathogens (e.g., candidemia, Group A streptococcal sepsis, staphylococcal sepsis), and when analyzed as a broad diagnostic category (Table 1). We postulated that the better outcomes in children reflect age-based differences in immune and inflammatory responses, possibly magnified by effects of more frequent co-morbidities in adults.

To better understand the basis for this childhood resistance, we began by identifying public datasets of transcriptome profiling performed on blood leukocyte samples in the high vs. low survival groups (children and adults, respectively). The analysis used Pathprint (Altschuler *et al*, 2013; Davis & Ragan, 2013; https://biocond uctor.org/packages/pathprint), a tool that is robust to batch effects and allows for comparison of gene expression at the pathway activity level across multiple array platforms. After identifying differences in pathway activity, we applied a novel method that is built upon the correlation of the expression of > 16,000 disease signatures from the Comparative Toxicogenomics Database (CTD), the Pharmacogenomics Knowledgebase (PharmGKB), pathway signatures from Wikipathways, KEGG, Netpath and Reactome, and drug signatures from CTD, PharmGKB, and CMap, across > 50,000 individual microarrays—the pathway drug network (PDN). The network neighborhood of the sepsis pathway signatures was used to identify the drugs that were most positively or negatively linked to high-survival (child) or high-mortality (adult) signatures. We assessed the validity of the top drug leads by analyzing prior data collected in preclinical animal models of sepsis and also by direct testing for improved survival in a mouse model of fatal endotoxemic shock.

# Results

## Key pathways differentiate the adult and child responses to sepsis

A total of 12 datasets reporting transcriptome profiling of whole blood samples from sepsis patients were identified for analysis from The Gene Expression Omnibus (GEO) and ArrayExpress databases (Barrett *et al*, 2013; Kolesnikov *et al*, 2015). The ultimate study population included 167 adults and 95 children, composed of 55 and 64% males, and mean ages of 59 and 8, respectively (Table 2). The Pathprint analysis tool was used to compare activity of pathways in adults and children with sepsis. Substantial differences in active or depressed pathways were identified, as illustrated in Fig 1. After applying thresholds based on the greatest age-associated differences, the four pathway clusters (A–D), detailed in Table 3, were used for further analysis. Tables EV1–EV3 provide additional details of Pathprint scoring and the results for all significantly different pathways.

## PDN base network: construction and benchmarking

The PDN methodology is a novel, pathway-centric drug discovery approach that tests whether an experimental gene signature is positively or negatively correlated to a gene signature associated with drug treatment. It relies on a base network constructed using the expression correlations between each of 16,150 drug, disease, and pathway gene signatures (collected from eight different databases), averaged across 58,475 publicly available human microarrays. By measuring the correlation between pathway, drug, and disease gene signatures over more than fifty thousand experiments, one can hypothesize whether the action that regulates, or is regulated by two signatures (e.g., a drug and a survival associated phenotype), may be linked and/or have similar actions (or opposing actions in the case of negative correlation). Since no comprehensive gold standard exists for evaluating the relationships between drug and disease signatures, to test the efficacy of the new PDN approach it was necessary to construct our own benchmark. Our benchmarking protocol involved the comparison of curated, known drug–disease relationships from the National Drug File Reference Terminology (NDFRT) and Structured Product Labels (SPL) databases (1,055 in total), with the drug–disease relationships produced by the PDN methodology. Beyond our goal of replicating the NDFRT and SPL drug–disease relationships using the PDN, we also compared our methodology with an alternative approach, Network Enrichment Analysis (NEA), a method based on gene-level curated protein–protein interactions (PPI; Alexeyenko *et al*, 2012). While both the PDN and PPI network approaches performed better than randomly assigning drug–disease relationships, the PDN decisively outperformed the PPI network at low false discovery rates (Fig EV1). Based on this benchmarking exercise, true-positive rates (TPRs) and false-positive rates (FPRs) were measured for the PDN and used to create a series of network cutoffs (the probability at which an edge is defined as true). From these analyses, a PDN cutoff parameter was chosen for the final base network that yielded as high as possible TPR (40%) while still keeping the FPR low (6%).

## PDN methodology results in high rates of positively validated drugs

Once the base network was constructed and subjected to benchmarking analysis, the next step was to challenge the network with a set of query pathways taken from our pre-defined Pathprint clusters A–D. Sub-networks of the PDN were constructed that contained these cluster pathways, together with their neighborhood of connected nodes. After several pruning steps (described in the Materials and Methods), the resulting network focuses on the gene signatures that relate most strongly to our cluster pathways. Through this

**Table 1. Epidemiological examples of childhood resistance to infectious and non-infectious injury.**

| Disorder | Child vs. adult difference | Child age range | Adult age range | Metric[a] | References |
|---|---|---|---|---|---|
| Historic data | | | | | |
| 1918 Pandemic flu | 176.2 vs. 786.5 | 5–14 | 20–34 | DP100K | Linder and Grove (1947) |
| Tuberculosis | 30.3 vs. 206.9 | 5–14 | 20–34 | DP100K | Linder and Grove (1947) |
| Measles | 0.05 vs. 0.5 | 5–15 | > 20 | CFR | Burnet (1952) |
| Yellow fever | 144 vs. 759 | 6–15 | 21–60 | DP100K | Canela Soler *et al* (2009) |
| Typhoid fever | 5 vs. 25 | 5–15 | > 20 | CFR | Burnet (1952) |
| Plague | 7 vs. 28 | 6–10 | > 16 | DR | Russell (1948) |
| Modern data | | | | | |
| Ebola | 57 vs. 81<br>60 vs. 72.5 | 5–15<br>5–15 | 20–60<br>> 16 | CFR | Rosello *et al* (2015)<br>Team *et al* (2015) |
| H1N1 2009 | 0.01 vs. 0.08<br>1.7 vs. 5.0 | 5–14<br>0–17 | 25–64<br>18–64 | DHR<br>DP100K | Van Kerkhove *et al* (2011)<br>Shrestha *et al* (2011) |
| Group A strep sepsis | 2.6 vs. 18 | < 13 | 19–96 | CFR | Megged *et al* (2006) |
| Staphylococcal sepsis | 2 vs. 25 | < 16 | > 16 | CFR | Denniston and Riordan (2006)<br>Laupland *et al* (2008) |
| Sepsis | 0.9 vs. 14.5 | 5–14 | 25–54 | DP100K | Melamed and Sorvillo (2009) |
| Sepsis (with co-morbidities) | 16.0 vs. 27.6 | 5–14 | 20–59 | CFR | Angus *et al* (2001) |
| Sepsis (without co-morbidities) | 6.3 vs. 12.8 | 5–14 | 20–59 | CFR | Angus *et al* (2001) |
| Severe malaria | 6.1 vs. 26.7 | ≤ 10 | 21–50 | CFR | Dondorp *et al* (2008) |
| Trauma (MOF) | 17 vs. 35 | < 16 | > 16 | CFR | Calkins *et al* (2002) |
| Acute chest syndrome (sickle cell) | 1.1 vs. 4.3 | < 20 | > 20 | CFR | Vichinsky *et al* (1997) |
| Candidemia | 10.1 vs. 30.2<br>15.8 vs. 30.6 | < 16<br>< 18 | ≥ 16<br>> 18 | CFR<br>CFR | Blyth *et al* (2009)<br>Zaoutis *et al* (2005) |
| Invasive pneumococcus infection | 3.8 vs. 21.5 | < 13 | 14–106 | CFR | Rahav *et al* (1997) |
| Chicken pox | 1.3 vs. 21.3<br>0.4 vs. 1.6 | 5–14<br>5–14 | ≥ 20<br>15–44 | CFR<br>CFR | Meyer *et al* (2000)<br>Joseph and Noah (1988) |
| Pneumonia | 2.5 vs. 9.4 | 5–14 | 20–64 | CFR | Tornheim *et al* (2007) |

This table shows the difference in mortality between children and adults for a variety of infectious diseases and types of injury. The age range identified as "child" or "adult" varied across the studies. When age was more narrowly stratified for children and adults, an average mortality rate was calculated based on the age ranges of 5–12 and 20–60, respectively.
[a]CFR, Case fatality rate; DP100K, Deaths per 100,000; DHR, Deaths to hospitalization ratio.

method, four network modules incorporating each of the Pathprint clusters A–D were created, containing 45 drug leads in total (Table 4).

This approach and other drug discovery methodologies generate enormous quantities of possible drug leads that necessitate efficient validation methods. Considering the large number of previous studies that have evaluated compounds for possible benefit in sepsis using animal models, we reasoned that one metric for evaluating the results from the PDN would be how often the identified drug leads corresponded to agents already shown to have positive (or negative) effects experimentally. Hence, we conducted extensive literature curation for each of the 45 compounds or closely related agents (e.g., ibuprofen for NSAIDs) and scored the presence of prior publications showing benefit or harm for survival in animal models of sepsis.

The validation efficacy of the drug list derived from Pathprint-to-PDN analysis was compared to three other gene-level drug discovery approaches as well as to a control approach (drugs selected at random from the entire list of CMap compounds). The first, a gene-level approach, also used PDN, but analyzed differentially expressed genes (DEGs) generated from a standard Limma analysis of children vs. adult transcriptomes, rather than pathway clusters (Appendix Fig S1). We found a substantially higher rate of positives in the list produced by a pathway-level PDN approach: 54%, compared to 27% for the gene-level approach, and 16% for randomly selected drugs. We also obtained up- and down-regulated DEGs from the BarCode method (McCall *et al*, 2010; Table EV4), an approach that categorizes gene expression as on or off, and used these genes, as well as the standard DEG list to query the LINCS database (Wang *et al*, 2016), a greatly expanded version of CMap (Lamb *et al*, 2006). The lists of compounds expected to have a positive effect on sepsis mortality (i.e., up- and down-regulated in adults compared to children) were also curated to assess the frequency of prior positive results in the literature. The percentage of positive drug leads achieved by the Pathprint-to-PDN methodology was significantly higher than with each of the four other methods ($P < 0.02$ by Fisher's exact test). The percent positives for each of the five categories of drug leads are summarized in Fig 2, and

**Table 2. Demographic information on datasets used for data-mining.**

| Study GSE no. | Age group | Age mean | Age range | Sex M | Sex F | Total | Time when sampled | Array GPL no. | References |
|---|---|---|---|---|---|---|---|---|---|
| 28750 | Adult | 60 | 38–82 | 6 | 4 | 10 | ≤ 24 h | 570 | Sutherland et al (2011) |
| 13015 | Adult | 55 | 40–81 | 11 | 18 | 29 | Time of diagnosis | 6947 | Pankla et al (2009) |
| 10474 | Adult | 58 | 18–83 | 18 | 16 | 34 | ≤ 48 h | 571 | Howrylak et al (2009) |
| 40586 | Adult | 59 | 37–75 | 8 | 7 | 15 | ≤ 48 h | 6244 | Lill et al (2013) |
| 57065 | Adult | 63 | 29–84 | 19 | 9 | 28 | ~ 30 min after onset shock | 570 | Cazalis et al (2014) |
| 33341 | Adult | 58 | 24–91 | 31 | 20 | 51 | Time of diagnosis | 571 | Ahn et al (2013) |
| 4607 | Child | 8 | 9–11 | 12 | 6 | 18 | ≤ 24 h | 570 | Cvijanovich et al (2008) |
| 9692 | Child | 7 | 5–9 | 6 | 2 | 8 | ≤ 24 h | 570 | Cvijanovich et al (2008) |
| 26440 | Child | 8 | 5–11 | 18 | 10 | 28 | ≤ 24 h | 570 | Wynn et al (2011) |
| 26378 | Child | 8 | 5–10 | 18 | 10 | 28 | ≤ 24 h | 570 | Wynn et al (2011) |
| 13904 | Child | 7 | 5–10 | 5 | 6 | 11 | ≤ 24 h | 570 | Wong et al (2009) |
| 40586 | Child | 8 | 7–8 | 2 | 0 | 2 | ≤ 48 h | 6244 | Lill et al (2013) |
| **Summary** | | | | | | | | | |
| | Adult | 59 | 18–91 | 93 | 74 | 167 | | | |
| | Child | 8 | 5–11 | 61 | 34 | 95 | | | |

The GEO database was queried to identify microarray transcriptome datasets from sepsis whole blood samples of adults and children. Samples from patients aged 18–91 comprised the adult group and patients aged 5–11 comprised the children's group. The table above specifies each study GSE no., age category, age mean, age range, the number of male or female patients, the timing of sample acquisition in the sepsis course, the array GPL no., and the reference used to access the original study.

details of the lists and references identified are provided in Appendix Tables S1–S5.

### PDN-derived therapeutic leads improve survival in murine endotoxemic shock

To directly investigate the utility of the PDN approach, we tested 10 of the top ranked compounds generated by the Pathprint-to-PDN method (Table EV5) for their effects on survival in an endotoxin shock model. Our goal was to use these drugs to directly modulate adult pathway signatures to match pathway signatures in children and potentially improve survival. Mice were pre-treated with the compounds as described in the Materials and Methods, followed by intraperitoneal administration of endotoxin. Five of the 10 compounds improved survival in this model (Fig 3). In all, eight of the 10 compounds had not been previously reported in sepsis survival studies; three of these eight showed benefits in our endotoxin shock model. The remaining two compounds were likely to be effective based on prior publications (topotecan, a water-soluble analog of camptothecin, and chlorpromazine, similar to piperacetazine) and they decreased mortality as expected.

## Discussion

In this study, we sought to address the dearth of effective drug treatments for sepsis by combining two novel approaches, summarized in Fig 4. Firstly, we focused on a remarkable natural experiment—the relative resistance to mortality in children vs. adults with sepsis. By data-mining publicly available whole blood transcriptomes, we

were able to identify key differences in pathway regulation between the two age groups. Continuing with a pathway-centric approach, we used pathway-based correlation to build a novel in silico drug discovery system to find drugs that might promote beneficial pathways (i.e., activated in children) or inhibit harmful ones (i.e., activated in adults). Evaluation of the resulting drug list by both curation and direct experimentation showed substantial enrichment for promising candidates.

The profiles of the five drugs found to be effective in vivo are diverse. Topotecan has broad anti-inflammatory effects attributed to inhibition of topoisomerase-dependent transcriptional activity of pathogen-induced genes (Rialdi et al, 2016). Chlorpromazine and amitriptyline share tricyclic structure and myriad potential mechanisms of action, e.g., interactions with neural receptors, but also inhibition of acid sphingomyelinase, which has been linked to decreased inflammation (Sakata et al, 2007). Vinpocetine is a synthetic derivative of the vinca alkaloid vincamine, with known anti-inflammatory properties (Jeon et al, 2010). Khellin is a folk medicine derived from the plant Ammi Visnaga and has a furanochrome structure, but benefits and mechanism(s) of action are poorly characterized. Although four of the five agents have anti-inflammatory properties, at least three of the five drugs that had no effect also have reported anti-inflammatory action [topiramate (Dudley et al, 2011), noscapine (Zughaier et al, 2010), and ethacrynic acid (Han et al, 2005)]. Additional investigation of these drugs (and other leads in addition to the top 10) may help identify meaningful commonalities more precisely.

By starting with human transcriptomic data in our comparison of children vs. adults, we substantially increased the potential value of subsequent analyses. Effectively, our approach uses human samples

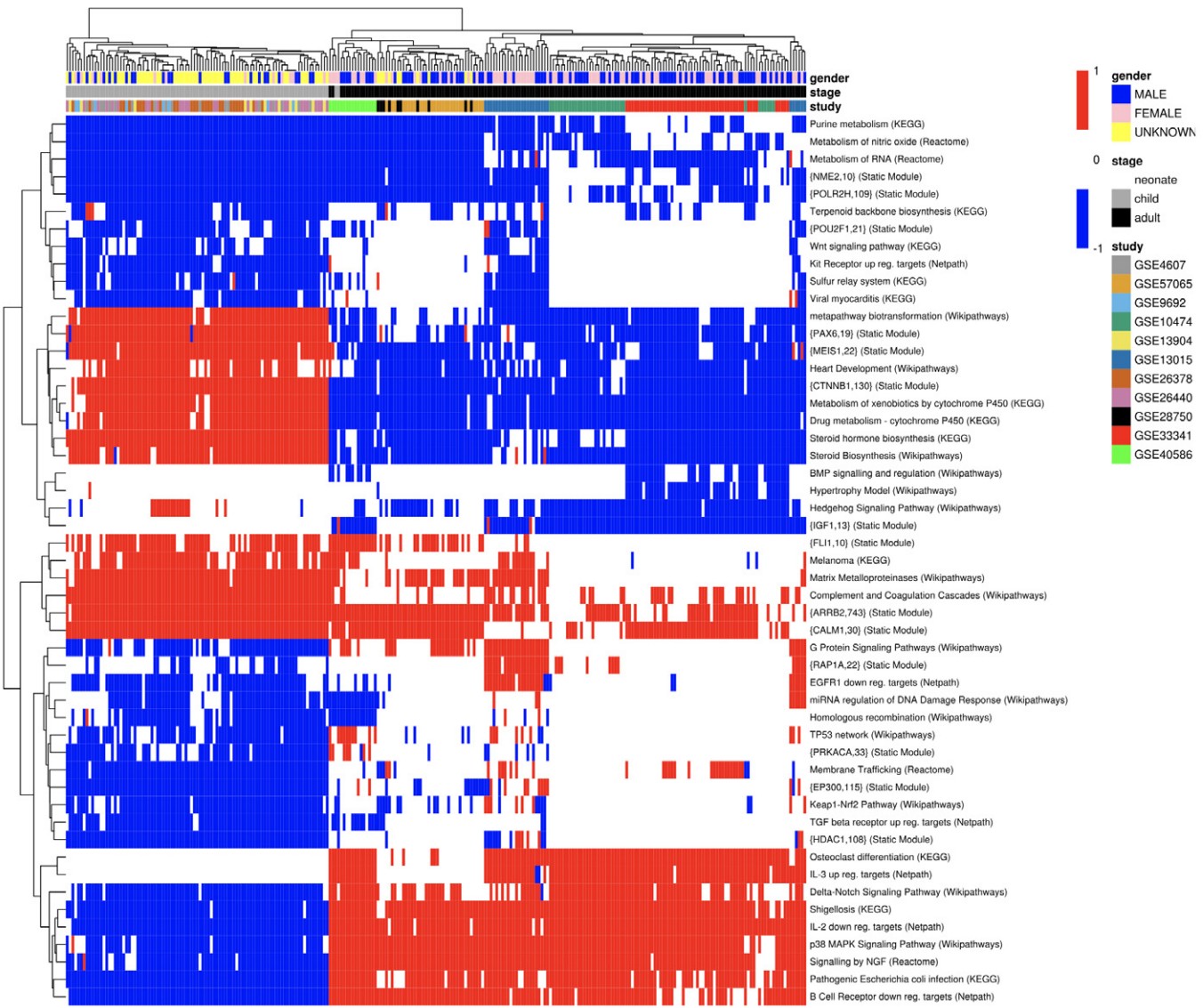

**Figure 1.  Sample heatmap generated from adult vs. child comparison using Pathprint.**

Pathprint analysis was used to analyze adult and child transcriptomes at the pathway level. To minimize intra-group variation and maximize inter-group variation, two filtering criteria were set in the generation of these data: (i) to maximize homogeneity within an age group based on minimizing the standard deviation, a cutoff of SD < 0.475 in the Pathprint score was used; (ii) to maximize differences between group comparisons using $t$-tests, Pathprint scores between groups were only included if $P < 10^{-10}$. The heatmap above was generated using the pheatmap package.

Source data are available online for this figure.

and then verifies the methodology in an animal model. The limitations of using animal models (especially mice) in preclinical sepsis studies are well recognized. Mice typically lack many of the common features of human sepsis patients (e.g., age, comorbidities, drug treatments, supportive care; Osuchowski *et al*, 2014; Efron *et al*, 2015) and exhibit highly species-specific transcriptomes after injury or sepsis (Seok *et al*, 2013). In addition, no model of sepsis in mice [e.g., endotoxemia, bacterial pneumonia, cecal ligation and puncture (CLP)] can completely replicate the physiological responses seen in human sepsis (Dejager *et al*, 2011). The strategy used here avoids reliance on an animal model of sepsis as the initial source of genetic information for the generation of a drug candidate list. Moreover, even the "gold standard" sepsis model, cecal ligation and puncture (CLP), is recognized as being technically difficult and variable; different responses are elicited from lab-to-lab or even from person-to-person within a given laboratory.

The (admittedly also imperfect) LPS model used in these studies, does fulfill an important criterion: It mirrors the pre- vs. post-pubertal human epidemiology that interests us, as detailed in our recent publication (Joachim *et al*, 2017). Thus, we believe that the endotoxemia model is a sufficient tool to begin our investigation of the underlying mechanisms driving pre-pubertal resistance.

**Table 3. Pathprint clusters chosen for drug candidate analysis using PDN.**

| | Children Pathprint score | Adults Pathprint score | Children-adults difference | *P*-value |
|---|---|---|---|---|
| **Cluster A (up in adults, down in children)** | | | | |
| IL-2 down reg. targets (Netpath) | −0.94 | 0.97 | −1.91 | 1.87E-88 |
| Shigellosis (KEGG) | −0.93 | 0.96 | −1.88 | 2.04E-88 |
| Endocytosis (KEGG) | −0.82 | 0.99 | −1.82 | 1.95E-56 |
| B cell receptor down reg. targets (Netpath) | −0.95 | 0.84 | −1.79 | 1.69E-109 |
| Signaling by NGF (Reactome) | −0.83 | 0.95 | −1.78 | 9.11E-66 |
| Pathogenic *Escherichia coli* infection (KEGG) | −0.96 | 0.82 | −1.78 | 1.51E-113 |
| Pentose Phosphate Pathway (Wikipathways) | −0.79 | 0.99 | −1.78 | 1.80E-50 |
| EGFR1 Signaling Pathway (Wikipathways) | −0.78 | 0.99 | −1.77 | 6.46E-57 |
| p38 MAPK Signaling Pathway (Wikipathways) | −0.80 | 0.95 | −1.75 | 2.91E-62 |
| {HCLS1,17} (Static Module) | −0.96 | 0.63 | −1.59 | 1.28E-64 |
| **Cluster B (down in adults, up in children)** | | | | |
| {CTNNB1,130} (Static Module) | 0.93 | −0.95 | 1.87 | 2.71E-91 |
| Metabolism of xenobiotics by cytochrome P450 (KEGG) | 0.86 | −0.98 | 1.84 | 2.55E-70 |
| Drug metabolism—cytochrome P450 (KEGG) | 0.84 | −0.96 | 1.81 | 7.02E-66 |
| Steroid hormone biosynthesis (KEGG) | 0.97 | −0.81 | 1.78 | 2.79E-128 |
| Steroid Biosynthesis (Wikipathways) | 0.87 | −0.89 | 1.77 | 3.90E-84 |
| **Cluster C (unchanged adults, down in children)** | | | | |
| {EP300,115} (Static Module) | −0.99 | −0.02 | −0.97 | 4.30E-75 |
| {HDAC1,108} (Static Module) | −0.99 | −0.02 | −0.97 | 2.46E-91 |
| Keap1-Nrf2 Pathway (Wikipathways) | −0.89 | −0.07 | −0.82 | 5.94E-48 |
| Kit receptor up reg. targets (Netpath) | −0.92 | −0.12 | −0.80 | 1.92E-52 |
| Sulfur relay system (KEGG) | −0.85 | −0.18 | −0.67 | 1.00E-29 |
| TGF beta receptor up reg. targets (Netpath) | −0.94 | −0.09 | −0.85 | 2.87E-67 |
| Viral myocarditis (KEGG) | −0.84 | −0.15 | −0.69 | 2.03E-32 |
| **Cluster D (unchanged adults, up in children)** | | | | |
| {FLI1,10} (Static Module) | 0.72 | 0.23 | 0.48 | 7.52E-15 |
| Melanoma (KEGG) | 0.77 | 0.12 | 0.65 | 8.98E-26 |
| Serotonin transporter activity (Wikipathways) | 0.72 | 0.22 | 0.49 | 1.73E-14 |
| Statin pathway (Wikipathways) | 0.96 | −0.08 | 1.04 | 4.30E-64 |

Four different clusters of pathways, generated through Pathprint analysis, were identified based on relative activation or inhibition in adults and children. The clusters were defined as follows: cluster A): expression up in adults, expression down in children; cluster B) expression down in adults, up in children; cluster C) expression unregulated (not significantly changed) in adults, down in children; cluster D) expression unregulated in adults, up in children. From each cluster, pathways showing the greatest divergence between the two age groups were selected for further analysis by PDN. This selection was based on a percentage (*N*) of samples that satisfied the criteria (*N* = 80% for clusters A–C; *N* = 70% for cluster D). More detailed descriptions can be found in Appendix Table S1.

Ultimately, we would like to expand the pre-pubertal resistance model to a species that is more similar to humans in sensitivity to endotoxin and sepsis—the rabbit. We especially note that a similar resistance to mortality from LPS in pre-pubertal vs. pubertal rabbits has been reported (Watson & Kim, 1963) although this finding was not the focus of the cited study.

Unfortunately, due to the absence of effective drugs for human sepsis, it is not possible to validate our method using human data. Therefore, we instead relied on outcomes in mice for both the *in vivo* testing (Fig 3) and the curation results (Table 4), which compiled drug treatment effects in studies mostly performed in murine models of sepsis. While imperfect, the

"reverse-translational" methodology (Efron *et al*, 2015) used in this work attempts to exploit the many remaining similarities in the murine and human responses to injury (Takao & Miyakawa, 2015). By limiting our study to pathways identified as important in humans, we diminish the risks of identifying murine-specific biology. Further assessment of the efficacy of the identified drug "hits" will need to be conducted in larger animal models and ultimately human patients. Despite the limitations, our approach offers a substantial improvement in isolating drugs that merit further evaluation in preclinical assays. The child vs. adult difference in the resistance to mortality may also prove useful as a starting point for drug discovery in other severe infections and disorders (Table 1). The change

**Table 4.  Curation of drug lists by literature search through PubMed.**

| Pathprint to drugs | Cluster | Prior data? | DEGs to drugs | Prior data? | Random | Prior data? |
|---|---|---|---|---|---|---|
| Fenoprofen | A | +/− | 0297417-0002B | | Urapidil | |
| Glibenclamide | A | + | Indomethacin | +/− | Trifluoperazine | |
| Asiaticoside | A | + | SB-202190 | + | Metaraminol | + |
| Topiramate | A | | Acetohexamide | + | Nomegestrol | |
| Suramin | A | +/− | STOCK1N-35215 | | Coralyne | |
| Hyoscyamine | A | + | Emetine | | Citicoline | |
| Pancuronium | A | | Tacrine | | Octopamine | |
| N-acetyl-L-leucine | A | | Thioridazine | | Sulfapyridine | |
| Mefenamic acid | A | + | Suloctidil | | Butoconazole | |
| Apigenin | A | + | Biotin | | 0175029-0000 | |
| Camptothecin | B | + | Cyclopenthiazide | | Tracazolate | |
| Lincomycin | B | + | Mebhydrolin | +/− | Tomatidine | |
| Ganciclovir | B | | Triprolidine | +/− | Tetroquinone | |
| Fursultiamine | B | | Colchicine | | Repaglinide | |
| Tocainide | B | + | Cinchonine | | Tiletamine | |
| GW-8510 | B | | Methoxamine | | Amikacin | + |
| Tanespimycin | B | + | Tanespimycin | + | Butirosin | |
| Carbenoxolone | B | + | Fluorometholone | +/− | Meptazinol | + |
| Tacrolimus | B | + | Nicardipine | + | Tolnaftate | |
| Conessine | B | | Quinpirole | | Fasudil | + |
| Khellin | C | | Cycloheximide | − | Enilconazole | |
| Eldeline | C | | Colchicine acid | | Sulfanilamide | |
| Sulfathiazole | C | | Meteneprost | − | Theophylline | |
| Geldanamycin | C | + | Puromycin | | Spiramycin | |
| Cefoxitin | C | | Digoxin | | Omeprazole | |
| Procaine | C | + | Naftidrofuryl | | Rolitetracycline | |
| Procyclidine | C | | Terfenadine | | Dexpropranolol | + |
| Monorden | C | + | Gelsemine | | Piribedil | |
| Hexetidine | C | | Sulindac | +/− | Sulfathiazole | |
| Piperacetazine | C | + | Drofenine | | Iobenguane | |
| Desipramine | A & C | | Thioguanine | | Dicycloverine | |
| Cyclosporine | A & C | + | Methylergometrine | | PF-0053978-00 | |
| Nifenazone | A & C | +/− | Methotrexate | | Dipivefrin | |
| Tanespimycin | A & C | + | Ethacrynic acid | | Aztreonam | + |
| Ethacrynic acid | D | | Dexamethasone | +/− | Tomatidine | |
| Noscapine | D | | Tolazoline | | Bicuculline | + |
| Tanespimycin | D | + | 3-aminobenzamide | + | Ethosuximide | |
| Mebhydrolin | D | | Epitiostanol | | Meclozine | |
| Vincamine | D | | Benzthiazide | | Alimemazine | |
| Altretamine | D | | 0179445-0000 | | Monensin | |
| Enalapril | D | + | Lidocaine | + | Prestwick-691 | |
| Coralyne | D | + | Alexidine | | Oxaprozin | +/− |
| Napelline | D | | Dihydroergocristine | | Amiodarone | |
| Clindamycin | D | + | Nifurtimox | | Ampicillin | |

A literature search using PubMed was performed to compare the number of therapeutic leads, generated by both pathway- and DEG-based drug prediction methods, which were shown to confer a survival benefit in *in vivo* mouse models of sepsis. Compounds were scored as follows: positive (prior studies showing survival benefit were identified: (+); both (prior studies showing both benefit and harm to survival were identified: (+/−); negative (prior studies showing only harm to survival were identified: (−); blank (no relevant studies were identified: no entry).

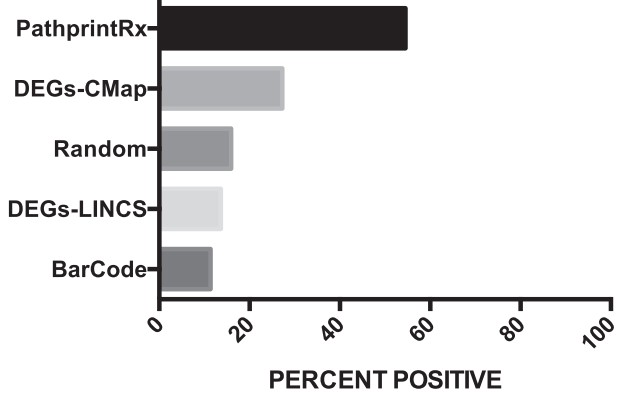

PERCENT POSITIVE

**Figure 2. Comparison of several methods of drug candidate identification.**

Five methods of transcriptome analysis/drug candidate identification were compared in their ability to successfully produce drug targets in at least one prior study showing a survival benefit from sepsis. (i) Pathprint-PDN: Comparison of pathways by Pathprint and drug candidate analysis by pathway drug network (PDN); (ii) DEGs-PDN: Comparison of differentially expressed genes (DEGs) by standard methods and drug candidate analysis by PDN; (iii) Random: Drugs chosen at random from the CMap database; (iv) DEGs-LINCS: Comparison of DEGs generated by standard methods and drug candidate analysis using LINCS database; and (v) BarCode-LINCS: Comparison of DEGs generated by BarCode method and drug candidate analysis using LINCS database. The three gene-level methods were found to be no better at generating positive drugs than picking drugs at random. All methods produced significantly lower percent positive rates than the Pathprint-PDN method ($P < 0.02$). Prism software (GraphPad) was used to compare the frequency of prior studies showing benefit for drug leads Fisher's exact tests.

in resistance is linked to the puberty transition, suggesting a role for sex hormones. Indeed, other experimental studies from our laboratory support this idea (Joachim *et al*, 2017; Suber & Kobzik, 2017) and indicate this topic merits further investigation in human studies.

In addition to the drug discovery goal of this work, the differences in pathway activation between adults and children also provide clues to the mechanisms driving childhood resistance to mortality. The initial pathway clusters generated through Pathprint were selected using relatively stringent criteria to maximize differences (see Materials and Methods). Using this approach, the pathways that were down-regulated in children in comparison to adults (Clusters A & C, see Table 3) involved response to infections (e.g., Shigellosis, Pathogenic *Escherichia coli* infection, viral myocarditis), canonical inflammatory and oxidative stress signaling pathways (e.g., IL-2 down-regulated targets, B cell Receptor down-regulated targets, p38 MAPK Signaling Pathway, Keap1-Nrf2 Pathway, TGF beta receptor up-regulated targets), pathways involved in growth and cell proliferation (e.g., Signaling by NGF, EGFR1 Signaling Pathway, Kit Receptor up-regulated targets), and pathways involved in chromatin modification. These pathways suggest a chronic up-regulation of the inflammatory response in adults in comparison with children. In general, there were fewer pathways that met our criteria for significant up-regulation in children in comparison to adults (Clusters B & D), and these were found to lack direct associations with inflammatory/immune responses. These pathways include lipid biosynthesis and regulation (e.g., Steroid hormone biosynthesis, Steroid

Biosynthesis, Statin pathway, cytochrome P450 activity), as well as proto-oncogenic genes and cancer (e.g., {CTNNB1, 130} [Static Module], {FLI1, 10} [Static Module], Melanoma [KEGG]). Using somewhat less stringent criteria, we identified the top 50 pathways (out of 633, ~ top 8%) that were up-regulated in adults but down-regulated in children or vice versa (ranked by the sum of their respective percentile ranks; Appendix Tables S1 and S2). The inflammatory (adult) vs. metabolic (child) difference is also evident in this comparison. The change in resistance is linked to the pubertal transition, suggesting a key role for sex hormones. Indeed, other experimental studies from our laboratory support this idea (Joachim *et al*, 2017; Suber & Kobzik, 2017) and indicate that this topic merits further investigation in human studies.

We were able to carry out the comparisons reported here due to the large number of datasets available that report whole blood transcriptomes in sepsis. This reflects the systemic nature of the condition, the accepted scientific importance of leukocytes in sepsis pathogenesis, and the relative ease of obtaining blood samples. However, whole blood transcriptomes have limitations. The expression profiles of whole blood essentially represent a weighted sum of the patterns of gene expression for each blood cell type and patients with sepsis exhibit heterogeneity in the leukocyte composition of the blood. No white blood cell count data were available in the data annotations for these studies, making us unable to directly control for these differences between individuals. However, the overall leukocyte differential in septic children in the 5–11 age range is very similar to that seen in adults (Stone *et al*, 1985; Park *et al*, 2014; Wong *et al*, 2015). Finally, the analysis of whole blood does not address potentially important contributions from endothelial, epithelial, tissue-resident immune, and parenchymal cell types (Cavaillon & Annane, 2006).

The novel drug development strategy applied here has more general applicability beyond sepsis. Classical approaches to understanding drug–disease relationships rely on experimental assays to relate cell states and perturbations to the etiology of different diseases, but cannot sample all possible interactions. Fully connected approaches such as the Molecular Signature Map (Ge, 2011) quantify interactions based on overlapping gene membership. This method successfully integrates our knowledge of gene lists but fails to address the issue of how drug, pathway, and disease signatures influence each other. We have used microarray data from the most highly represented platforms in GEO to determine the correlation of the expression of over 16,000 drugs, diseases, and pathway gene signatures in humans. In constructing the PDN, we used partial correlations in order to quantify the relationship between network nodes while still accounting for the influence of the other gene signatures. The resulting network enables us to interpret the cell as a whole based on the relationships and flow of information among the myriad processes occurring within it.

Prior to the development of the PDN methodology, the steps used in the CMap pipeline have been the main transcriptome-based drug discovery paradigm. The standard CMap pipeline tests whether an experimentally derived up- and down-regulated gene signature is also up- or down-regulated in a set of drug perturbation expression data. Broadly, this is equivalent to querying whether the transcriptional impact of the experiment is similar, or opposite to the transcriptional impact of a drug in CMap. Our alternative approach tests whether an experimental gene signature is correlated or

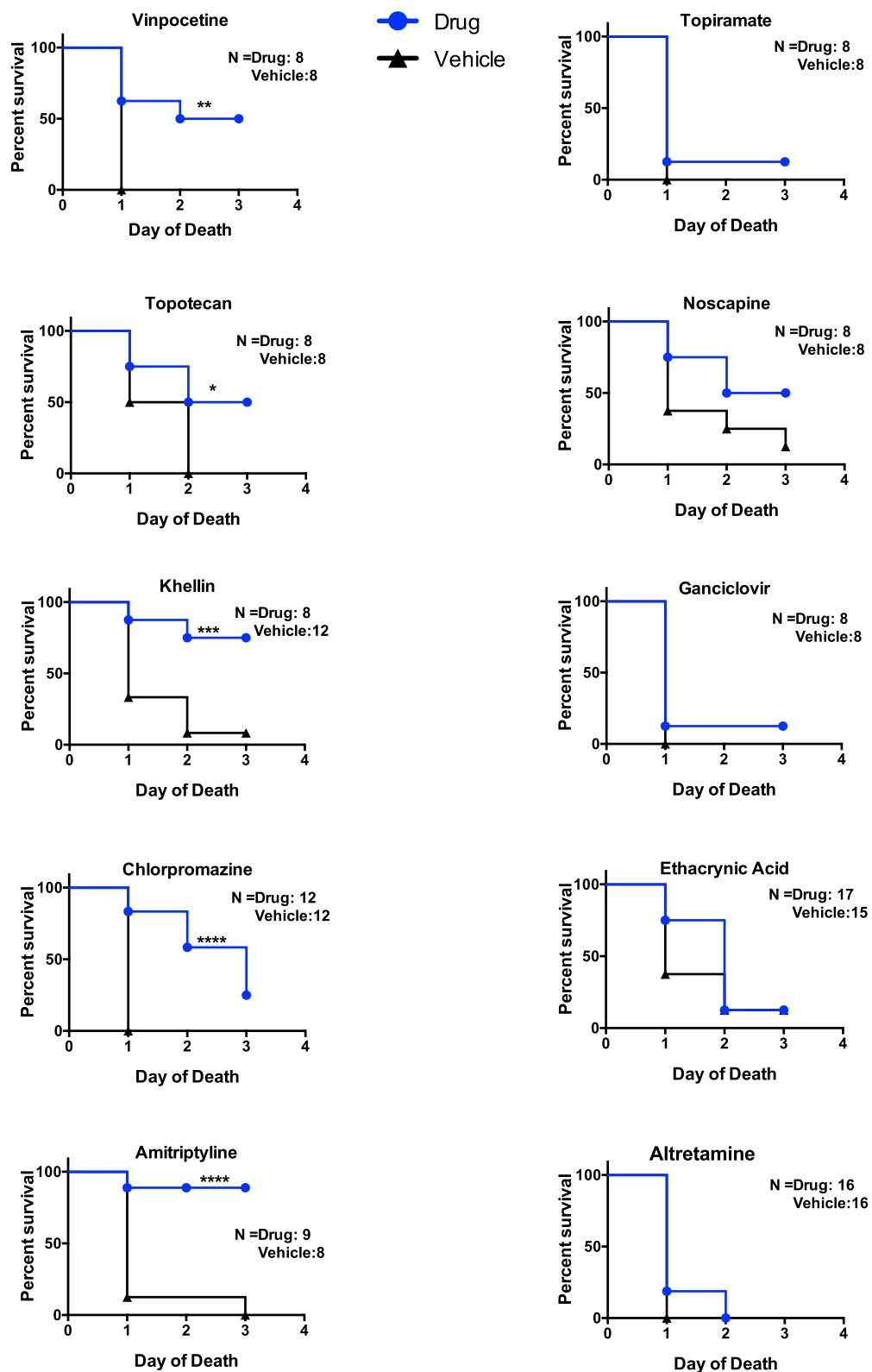

**Figure 3.  Validation of select PDN drug candidates in an *in vivo* endotoxemia model.**

Therapeutic leads generated using PDN were directly tested for survival benefit using a murine model of endotoxemia. Select compounds were injected 24 h before and on the day of LPS administration, using routes and doses specified in the methods. C57bl/6 female mice were injected with a high-lethality dose of *Escherichia coli* LPS (38–40 µg/g) followed by a subcutaneous injection of sterile saline. Significant differences in concentration between drug and vehicle-treated pre- and post-pubertal mice are labeled with ****$P < 0.0001$, ***$P < 0.001$, **$P < 0.01$, or *$P < 0.05$. Percent survival was compared using a log-rank Mantel–Cox test.

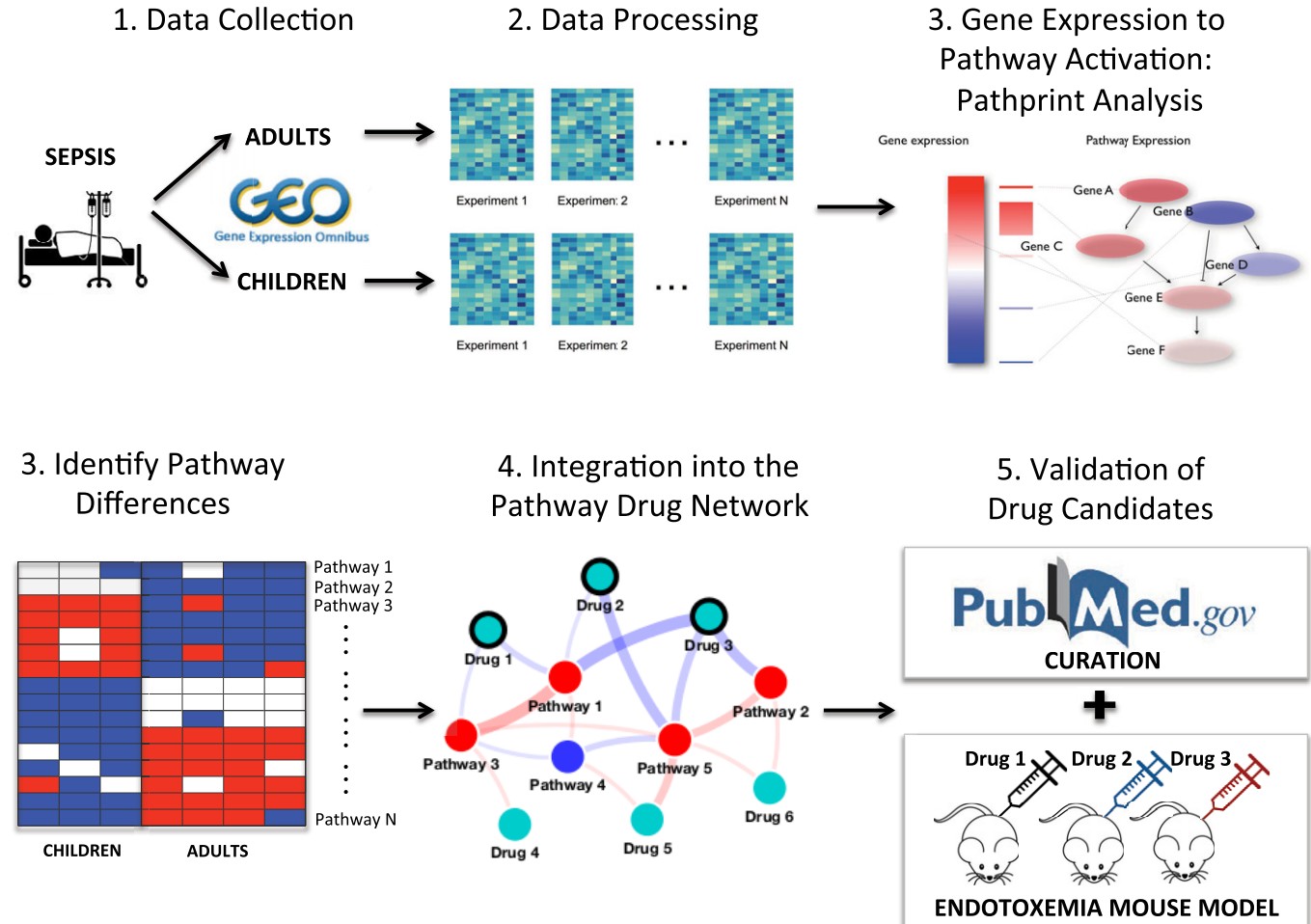

**Figure 4.   Summary workflow.**

We began by identifying publicly available datasets from transcriptome profiling experiments that analyzed blood leukocyte samples from adult and child sepsis patients. After data processing, we used Pathprint to translate these gene expression patterns to the pathway activity level. By comparing samples at the pathway level, the Pathprint method is robust to batch effect and allows for comparison across multiple array platforms. After identifying age-associated differences in pathway activity, we used them to facilitate drug discovery by constructing targeted pathway drug networks (PDNs). This novel method works by incorporating our target pathways into a base network built upon the correlation in the expression of > 16,000 disease, pathway, and drug gene signatures across > 50,000 individual microarrays. The resultant network neighborhood was used to identify drugs with positive or negative association with high-survival (child) or high-mortality (adult) pathways, respectively. We validated top drug leads by curating and analyzing prior data collected in preclinical models of sepsis and also by directly testing their ability to improve survival in a mouse model of fatal endotoxemia.

anti-correlated to the gene signature associated with drug treatment. Importantly, the correlation is measured not just in the setting of the transcriptome data from a single experiment, but across many experiments (over 50,000 arrays from over 2,000 experiments). The rationale for the PDN methodology is to quantify the relationship between two signatures across many experiments rather than assessing their similarity in a single test. If correlation is detected, we can hypothesize that the action that regulates, or is regulated by, those two signatures [i.e., from the drug and from the experimental phenotype (e.g., better survival)] may be linked and/or have a similar action (or opposing action in the case of negative correlation).

Our approach is not meant to directly replace CMap, but to greatly expand its power by exploiting experimental linkage. CMap can take any signature as an input, but the pre-defined array sets upon which is tested are fixed, and also limited in scope to experimentally testable perturbations. The core focus and strength of the

PDN methodology is the ability to link any pair of gene signatures in terms of their transcriptional regulation, irrespective of their source. We have included a range of additional gene signatures in the analysis so that the links are not restricted to drug interactions. In addition, the output of the method is a network, rather than a list of pair-wise relationships, meaning that clusters of drugs can be detected along with any closely associated pathways. The long-term goal is to make use of the relationships between the drug, pathway, and diseases signatures in the network to suggest mechanisms of action for drug leads. The eventual aim is to link pathways, drugs, mutations, and diseases all based on the same background dataset.

Limitations of this approach include some of the well-recognized problems in meta-analysis of microarray data in general (Tseng *et al*, 2012) and in sepsis specifically (Fiusa *et al*, 2014; Sweeney & Khatri, 2016). The Pathprint approach overcomes some of the problems in merging data from different platforms. However, because it

cannot integrate all platforms, some sepsis studies could not be included. The study relied on very useful, but imperfect databases. For example, the extensive reliance of CMap (and LINCS) on cancer cell lines may skew results. The background data used for the calculation of the PDN correlations will be subject to a similar investigation bias in the samples uploaded to the GEO database. It is also important to note that the quantity of available transcriptomic data (microarrays & RNA-Seq datasets) has grown (and continues to expand daily) since the PDN base network was first constructed. A rich collection of other approaches to data-mining exists in the literature (Pathan *et al*, 2015; Henriques *et al*, 2017; Li *et al*, 2017). Integration of other analytic strategies might offer additional insights, and this exploration merits future consideration. Similarly, an expanded PDN based on a current version of the LINCS database (now accessed at clue.io) might provide additional power.

Indeed, the overall success rate of drugs identified by "reversal of signature" methods is unknown, but supported by individual successes (Iorio *et al*, 2013; Musa *et al*, 2017). A further limitation of existing, pair-wise approaches to determine drug–disease relationships, including that approach presented here, is that no mechanistic data can be inferred. The integration of pathway and experimental gene signatures in the network allows for the identification of tightly connected pathway sub-networks around each drug–disease connection. Furthermore, the network allows for both negative and positive connections to be identified, significantly distinguishing this approach from existing overlap-based *in silico* methods. These features improve the identification of drugs with synergistic effects or sets of drugs with independent mechanisms of action on a disease, factors that are vitally important in overcoming polygenic drug resistance.

The ultimate aim of this work was to discover novel drug candidates for the treatment of sepsis by data-mining and comparing whole blood transcriptomes from two populations with naturally high (adults) or low (children) susceptibility to death from sepsis. Pathways with age-specific activation were identified through Pathprint and successfully used to interrogate the pathway drug network (PDN), which allowed for the identification of medications that could promote beneficial pathways during sepsis (i.e., activated in children) or inhibit harmful ones (i.e., activated in adults). Validation by literature curation and direct experimentation in endotoxemic mice indicated that the resulting drug list contained many promising therapeutic candidates. These findings suggest that our unique, pathway-centric approach to drug discovery may prove a powerful new tool in identifying novel therapeutics for sepsis and other complex medical conditions.

# Materials and Methods

### Study design

The objective of this study was to collect publicly available whole blood transcriptomes from septic adults and children, and then employ pathway-based bioinformatics tools to identify differentially regulated pathways and discover novel drug candidates for the treatment of sepsis. The GEO and ArrayExpress databases (Barrett *et al*, 2013; Kolesnikov *et al*, 2015) were used to identify publicly available microarray transcriptome datasets from whole blood samples of septic patients. The criteria for inclusion of microarrays were (i)

the availability of annotation data for the age of subjects, (ii) the use of microarray platforms supported by the Pathprint tool, and (iii) satisfactory evaluation by quality control analysis.

Pathways with age-specific activation were identified through Pathprint and used to interrogate the base PDN. The reliability of the PDN in its ability to identify accurate drug-disease relationships was benchmarked against known, curated relationships from the NDFRT and SPL databases. Further validation of the drugs identified through the PDN methodology was performed through literature curation as well as direct experimentation in endotoxemic mice.

All validation experiments using mice were conducted in strict adherence with the NIH Guide for the Care and Use of Laboratory Animals. The number of mice was chosen based on past success in evaluating interventions to improve survival in infectious disease models. Endpoints in these studies were survival (for over 72 h) or mortality. Analysis of mortality included counting deceased mice as well as humane euthanasia of mice with severe, pre-terminal morbidity.

### Transcriptome data processing

GEO and ArrayExpress databases were queried to identify microarray transcriptome datasets from sepsis whole blood samples. Samples from patients aged 5–11 comprised the children's group while samples from patients 18 years of age or older comprised the adult group (details of demographics and datasets used are provided in Table 2). The age range for the children's group was chosen because it is similar to the 5–14 age group that showed greater survival in the 1918 influenza pandemic (Ahmed *et al*, 2007), but adjusted to reflect the earlier onset of puberty in modern times (Ong *et al*, 2006; Toppari & Juul, 2010).

The main workflow began by curating datasets and metadata of interest. This curation process involved both automated steps (e.g., database searches for keywords) as well as manual work to compile and identify whether the required metadata were available (e.g., age of subject providing sample in a given dataset). In some cases, the authors of the individual studies were contacted to obtain such information. The retrieved metadata were filtered and standardized and the relevant annotations of interest extracted (i.e., age, sepsis status, gender, data locations). The curated metadata from each study were then combined to create a covariate table that was used to download each sample's expression data using the GEOquery package in R (Davis & Meltzer, 2007). To analyze data from multiple array platforms, differential activation of pathways was assessed using the R package Pathprint. To identify pathways with minimal intra-group variation and maximal inter-group variation, two filtering criteria were set: (i) to maximize homogeneity within an age group based on minimizing the standard deviation (a cutoff of SD < 0.475 in the Pathprint score was used); (ii) to maximize differences between group comparisons using *t*-tests (Pathprint scores between groups were only included if $P < 10^{-10}$). Heatmaps to visualize differences in pathway activation were generated using the pheatmap package (cran.r-project.org/web/packages/pheatmap/index.html).

A gene-level analysis was also performed on the subset of datasets that used the same array (Affymetrix HG-U133 Plus 2.0, GEO accession GPL570) as follows. First, quality control and normalization were performed using the arrayQualityMetrics (Kauffmann

*et al*, 2009) and RMA packages (Irizarry *et al*, 2003). DEGs were identified using Limma (Ritchie *et al*, 2015). The problem of batch effects in gene expression analysis is well known (Leek *et al*, 2010; Lazar *et al*, 2012). Pathprint addresses this issue by aggregating expression at the level of a pre-defined pathway. In contrast, an earlier methodology called the Gene Expression Barcode (McCall *et al*, 2011) operates at the level of the gene. To allow for comparison of results, the Barcode method was applied using the fRMA package (McCall *et al*, 2010). Filtering by binary entropy measures (< 0.295 for intra-group binary entropy and > 0.3 for inter-group binary entropy) was used to identify genes with maximal expression differences between age groups. The top up- and down-regulated genes in the adult vs. child comparisons were used to query the LINCS database (Duan *et al*, 2014). Using the percentile rank ("mean_rankpt_2"), the top 45 compounds anti-correlated to the adult profile were selected and subsequently evaluated for published evidence of efficacy in sepsis models as described below.

### Selection of pathways for analysis by PDN

Four different clusters of pathways, generated through Pathprint analysis, were identified based on similar patterns of relative expression: cluster A) expression up in adults, down in children; cluster B) expression down in adults, up in children; cluster C) expression unregulated (not significantly changed) in adults, down in children; cluster D) expression unregulated in adults, up in children. Pathways from each cluster that showed the greatest difference between the two comparison groups were selected for further PDN analysis. This selection was based on the percentage (*N*) of samples that satisfied the criteria (*N* = 80% for clusters A–C; *N* = 70% for cluster D). For example, the pathways selected from cluster A were up-regulated in at least 80% of samples in adults AND down-regulated in at least 80% of samples in children. For cluster D, use of the 80% criterion produced only one pathway for evaluation. Hence, the threshold was lowered to allow inclusion of the four pathways that were up-regulated in at least 70% of the samples from children and unregulated in adults.

### Curation of gene-sets for PDN base network creation

A set of drug, disease, and pathway gene-sets were curated from the following resources:

1   Comparative Toxicogenomics Database (CTD) (2,452 chemical/drug and 609 disease gene-sets): The CTD (Davis *et al*, 2017) includes curated data describing cross-species interactions between chemicals and genes/proteins as well associations between chemicals, genes, and diseases. The data were retrieved from the CTD, MDI Biological Laboratory, Salisbury Cove, Maine, and NC State University, Raleigh, North Carolina (http://ctdbase.org/) [5 November 2012 retrieval].

2   The Pharmacogenomics Knowledgebase (PharmGKB) (178 chemical/drug gene-sets, 78 disease gene-sets): PharmGKB (Whirl-Carrillo *et al*, 2012) is a pharmacogenomics knowledge resource that encompasses clinical information including dosing guidelines and drug labels, potentially clinically actionable gene–drug associations and genotype–phenotype relationships. Data (updated 11/6/12) were downloaded from the PharmGKB website (www.pharmgkb.org).

3   Connectivity Map (CMap) (12,200 chemical/drug gene-sets): CMap (Lamb *et al*, 2006) is a collection of genome-wide transcriptional expression data from cultured human cells treated with bioactive small molecules. CMap contains 6,100 expression profiles representing 1,309 compounds. The data can be retrieved from http://www.broadinstitute.org/cmap. The rank matrix available on the website (contains 22,283 gene probes and 6,100 samples) was used to build unique gene signatures for each perturbation (drug treatment). Probe sets were ranked in descending order of the ratio of the treatment-to-control values. The probe that was most up-regulated relative to the control was designated as top rank (#1), while the probe that was most down-regulated relative to the control was designated as bottom rank (#22,283). Separate up- and down-regulated gene signatures in response to each drug were compiled using the top and bottom 1% of ranked genes, respectively. These gene signatures served as a proxy for the transcriptional impact of a drug and allowed for the addition of CMap nodes to the PDN.

4   Pathprint (633 gene-sets): Gene-sets from the pathways used by the Pathway Fingerprint (Pathprint; Altschuler *et al*, 2013) were taken from the R package Pathprint (compbio.sph.harvard.edu/hidelab/pathprint/Pathprint.html) implemented in Bioconductor (bioconductor.org/packages/pathprint/). The pathway list contains gene-sets derived from a range of pathway databases (Reactome, KEGG, Wikipathways, Netpath; see Pathway Fingerprint for description and references), and modules derived from a functional gene interaction network known as "static modules" (Wu *et al*, 2010).

The gene-sets derived from each of the resources described above were combined to create a library of 16,150 unique gene signatures.

### PDN base network construction

A base network was constructed using the correlation between the expression levels of each of the 16,150 signatures, across 58,475 publicly available human microarrays (Affymetrix HGU133 Plus2) obtained from GEO. The array set contained 2,120 experiments, the same set of microarrays that make up the GPL570 expression background in the Pathprint package (see Bioconductor package for full list). For each microarray, the genes were ranked by expression level, from #1 (low expression) to T (high expression), where T is the total number of genes in the array. The expression score, En(G), for a gene signature, G, of size k, represented in an array by genes $g_1, g_2 \ldots g_k$, is defined by the mean squared rank of the member genes, $En(G) = n^{-1} \times \sum R_i^2$, where $R_i$ is the rank of gene $g_i$ in a pathway containing $n$ genes. The network edges are represented by the partial correlation between each gene signature expression score, which is the correlation coefficient between two gene signature expression scores after accounting for the influence of the other gene signatures. The partial correlation was calculated using the R package GeneNet, which makes use of shrinkage estimators of partial correlation for fast and statistically efficient processing of the data (cran.r-project.org/web/packages/GeneNet/index.html).

The significance of each of the connecting edges was assessed by fitting a mixture model to the partial correlations, where the null model is estimated from the data. The calculation used the R

package, fdrtool (http://cran.r-project.org/web/packages/fdrtool/index.html) to generate two-sided *P*-values for the test of non-zero correlation for each edge, corresponding posterior probabilities for edges, and *q*-values (Schafer & Strimmer, 2005). The PDN method creates a network that is dynamic and can be extended to cover any number of additional signatures. The network was benchmarked using curated case–control interactions.

## Network characterization

### PDN network topology

The PDN degree distribution and degree cumulative probability are shown in Fig EV2. At first glance, the degree distribution of the PDN plotted on a log-log scale may be considered roughly linear—indicative of a scale-free network following a power law distribution with gamma of approximately 0.61. However, the cumulative probability plot, which would also be linear on a log-log scale under scale-free conditions, clearly demonstrates significant divergence from a power law distribution. An exponential distribution or power law with exponential cutoff provides more reasonable but not exact fits. Scale-free networks were thought to be a common characteristic of biological networks (Albert, 2005), generally rationalized by the hypothesis that such networks are robust to random breaks and facilitate rapid inter-node communication by short average path lengths and high clustering coefficients. However, as higher resolution experimental data have become available, the general scale-free nature of biological networks has been increasingly questioned (Lima-Mendez & van Helden, 2009). The degree distribution of the PDN is evidence of a denser structure than would be expected from a power-law distributed network and provides an additional example of departure from scale-free topology. It should be noted that the presence of the drug perturbation and disease-state signatures in the PDN would also be expected to disrupt the structural characteristics of a network based on canonical pathways alone.

### Biological interpretation of PDN pathway relationships

We wished to establish whether pathways are correlated within the PDN in biologically meaningful ways. We created a sub-network consisting of all pathways from the network. Markov Clustering (van Dongen, 2000) of pathway–pathway correlation using the Graphia Pro environment (Kajeka.com) generated 38 biologically consistent clusters containing between 6 and 34 pathways as nodes (Table EV6). Pathway nodes vary in degree from 244 (Pathway.{PRKACA,33} (StaticModule)) to 4 [Pathway.TNF-alpha/NF-kB Signaling Pathway (Wikipathways)]. Interpretation of cluster membership is complicated by the fact that only a partial understanding of known functional relationships between pathways exists. We have begun to address this challenge in a separate study of global pathway relationships (Pita-Juarez *et al*, 2018).

Clusters show pathways related by function. As cluster size decreases, functions become more specific. One example of this can be seen in our largest cluster, Cluster 1, which contains pathways sharing functionality across cellular responses to stress, infections and cancers, B and T Cell receptor signaling pathways, as well as Tuberculosis, Leishmaniasis, and Toxoplasmosis pathways. Other clusters are enriched for pathways in lipid metabolism (Cluster 3); cell cycle and DNA replication (Cluster 6); immune signaling (Cluster 8); DNA repair and replication and RAS family genes (Cluster

10); extracellular matrix function (Cluster 12); and electron transport, respiratory chain function, oxidative phosphorylation, and Parkinson's disease (Cluster 14). Functional relationships between pathways structured in this clustering approach may be insightful in terms of providing data driven context to known relationships. For instance, shared functionality in immune-mediated mechanisms of stress surveillance in cancer is an existing observation (Seelige *et al*, 2018).

## Benchmarking the PDN

In an effort to benchmark the PDN, we compiled two sets of curated drug–disease relationships: 149 documented relationships from the NDFRT database (46 diseases and 92 drugs) and 906 documented relationships from the SPL database (58 diseases and 122 drugs). The drug and disease terms from both of these databases have been previously mapped to the PharmGKB identifiers (Zhu *et al*, 2013) used in construction of the PDN. This allowed for direct comparison of the two methods. CMap datasets have not been mapped to the terms in the NDFRT and SPL databases and were thus not used for benchmarking. True- and false-positive rates (TPRs, FPRs) were measured for the PDN and used to create a series of network cutoffs, and an ROC curve was plotted (Fig EV1). Beyond the goal of replicating these drug–disease relationships using the PDN, we also compared our methodology with an alternative approach, NEA.

## Determining CMap drugs associated with query cluster pathways

Once the base network was constructed, we interrogated it with a set of query pathways from pre-defined Pathprint clusters A–D. A sub-network was constructed for each cluster (e.g., Cluster A) that contained the nodes representing each of the member pathways of that cluster, together with all base network nodes with connecting edges to the cluster members. To assure that the new network was specific to correlations associated with a given cluster of pathways, we further pruned the sub-networks by removing base network nodes if they did not connect to at least three or more of the pathways in the cluster. Next, we ranked the significance of the remaining base nodes using the *P*-values of the edges connecting them to each of the cluster nodes, aggregated by Fisher's method. For all non-CMap nodes, *P*-values were simply aggregated across the entire pathway cluster into a single *P*-value. For CMap nodes, *P*-values were first aggregated across the separate CMap up- and down-regulated gene signatures for each drug and secondly across the entire pathway cluster. An overall positive correlation between a cluster pathway and a pair of CMap nodes was determined by combining the *P*-values calculated for positive correlation with the up-regulated CMap drug signature and negative correlation with the corresponding down-regulated CMap drug signature. Overall, negative correlation between a cluster pathway and a pair of CMap nodes was established in a similar way. The *P*-values for positive and negative correlation were then ranked and combined into a simple combined association score: Score = rank(negativeRank − positiveRank)/(nDrugs/2) − 1, where negativeRank is the rank of the negative *P*-value, positiveRank is the rank of the positive *P*-value, and nDrugs is the number of drugs tested. Any CMap drug with a *P*-value of > 0.1 for both positive and negative association was given a score of 0. Thus, a negative score means that the drug

opposes the activity of the cluster pathways, a positive score means that drug enhances the activity of the cluster pathways. A score of 0 means no significant interaction. The 10 highest negatively scoring drugs each for clusters A and C, the 10 highest positively scoring drugs each for clusters B and D, as well as the top 5 within the overlap of clusters A and C (a total of 45 drugs) were prioritized for validation as described in the following section. While the PharmGKB database also contained drug signatures, they were non-directional (up and down-regulated genes are not distinguished). Because it would have been impossible to distinguish a correlated vs. anti-correlated signature using the PharmGKB signature, we chose to perform all final analyses using the CMap signatures.

### Determining CMap drugs associated with an individual gene signature

To compare PDN functionality based on a network analysis of a target cluster of pathways with standard single gene list-based analysis, we queried the PDN directly with a differential signature derived from adult and child transcriptomes (see above). The top 500 up- and down- regulated probes from a comparison of children vs. adults using datasets limited to a single array platform (GPL570) were matched to 427 and 405 up- and down-regulated genes in CMap. These gene signatures were incorporated into the PDN and ranked for positive or negative association with CMap drug signatures by a similar approach as the Pathprint cluster pathways described above. The differentially expressed genes were split into up-regulated and down-regulated gene-sets. These gene-sets were then introduced into the PDN as two new nodes and evaluated separately. The *P*-values of the edges connecting each of the new nodes to the separate CMap up- and down-regulated gene signatures were aggregated for each drug. This calculation helped to quantify whether the up- and down-regulated components of the adult vs. child differentially expressed signatures were positively or negatively associated with each CMap drug. Then, an overall positive or negative correlation between a differential gene signature and a CMap drug was determined by combining the *P*-values calculated for the up-regulated and down-regulated parts of the signature. The top 45 negatively associated drugs were selected for further validation.

### Validation of drug leads

To validate the top 45 therapeutic leads generated by each drug discovery methodology, a literature search using PubMed was performed. We used terms (i.e., keywords: survival, mortality, sepsis, endotoxin) to identify studies that tested a particular drug, or a closely related compound, for *in vivo* benefit in animal models of sepsis. Compounds were scored as follows: positive (prior studies showing survival benefit were identified); both (prior studies showing both benefit and harm to survival were identified); negative (prior studies showing only harm to survival were identified); no score was assigned when no relevant studies were identified. The efficacy of the Pathprint-to-PDN methodology was compared to several other transcriptome-to-drug discovery approaches (described in results and Fig 2) as well as to drugs randomly chosen from the CMap database.

Ten drugs were selected from the pool of drugs identified using the Pathprint-to-PDN methodology. These were chosen to sample from all clusters and to include agents both with and without prior evidence of potential benefit. Substitutions with highly similar compounds were made for some of the predicted drugs. For three of the four substitutions made (topotecan, chlorpromazine, amitriptyline), the rationale was driven by the existence of publications showing survival benefit in animal models using the substituted drug ((Brand *et al*, 2008; Rialdi *et al*, 2016; Villa *et al*, 1995), respectively). These data did not exist for the original predicted drugs. For the final pair, vincamine/vinpocetine, there are no published data demonstrating a survival benefit, but there are data showing that vinpocetine has some anti-inflammatory activity (Jeon *et al*, 2010). No similar data were found for vincamine, thus motivating our choice of vinpocetine.

The compounds were directly tested for effects on survival using a murine model of endotoxemia. These experiments were conducted in strict adherence to the NIH Guide for the Care and Use of Laboratory Animals, and under a protocol approved by the Harvard Medical Area Institutional Animal Care and Use Committee (IACUC). C57bl/6 female mice (5 weeks old, Charles River, Wilmington, DE) were injected with a high-lethality dose (e.g., 38–40 μg/g) of *E. coli* LPS (L3755; Lot: 123M4096V; Sigma-Aldrich, St. Louis, MO, USA) between hours 5 and 7 of the light period in the animal facility (12–2 PM). In order to mitigate fluid loss and dehydration, each mouse was also given a subcutaneous injection of sterile saline (equal to 2.5 % of body weight). To test the effects of drug leads, compounds were injected 24 h before and on the day of LPS administration, using routes and doses specified in Table EV7. Analysis of mortality included counting deceased mice as well as humane euthanasia of mice with severe, pre-terminal morbidity (scored by evaluation of appearance, movement, and response to touch).

### Statistical analysis

The statistical methods used in transcriptome comparisons as well as the creation and application of the PDN methodology are detailed in the sections above. Fisher's exact test was used to compare the frequency of prior studies showing benefit for drug leads across the multiple transcriptome-to-drug methodologies. A log-rank (Mantel–Cox) test was used to analyze murine endotoxemia survival data. Both of these statistical analyses were performed using Prism software (GraphPad, San Diego, CA).

**Expanded View** for this article is available online.

### Acknowledgements
This work was supported by NIH ES00002, NIH 5T32HL007118, DARPA W911NF-10-1-0217, a CIRCUITS grant from the Cure Alzheimer's Fund, the Joseph D. Brain Fellowship Fund, the Jere Meade Fellowship Fund, and the National Institute for Health Research (NIHR) Sheffield Biomedical Research Centre (Translational Neuroscience)/NIHR Sheffield Clinical Research Facility.

### Author contributions
Study conception and design: RBJ, GMA, WAH, LK; data-mining, annotation and analysis: RBJ, GMA, JNH, HRW, WAH, LK; drafted manuscript: RBJ, GMA, WAH, LK; and critical revision: RBJ, GMA, HRW, WAH, LK.

### Conflict of interest
The authors declare that they have no conflict of interest.

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
