## [Review Process File · Molecular Systems Biology]

The Relative Resistance of Children to Sepsis Mortality: From Pathways to Drug Candidates

Rose B. Joachim, Gabriel M. Altschuler, John N. Hutchinson, Hector R. Wong, Winston A. Hide & Lester Kobzik

Review timeline:

Submission date:	6 October 2017
Editorial Decision:	8 December 2017
Revision received:	2 March 2018
Editorial Decision:	17 April 2018
Revision received:	20 April 2018
Accepted:	25 April 2018

Editor: Maria Polychonidou

Transaction Report:

1st Editorial Decision

8 December 2017

Thank you again for submitting your work to Molecular Systems Biology. As you will see below, both reviewers appreciate that the presented findings seem interesting. They raise however a series of concerns, which we would ask you to address in a major revision.

The reviewers' recommendations are rather clear so I think that there is no need to repeat all the points listed below. Reviewer #2, whose primary expertise is computational biology, mostly raises concerns regarding the approach for inferring drug targets based on the analysis of transcriptome data. These concerns need to be addressed since the PDN approach is an integral part of the study. Reviewer #3 focuses on aspects related to the follow up analyses and the therapeutic relevance of the findings. S/he makes several constructive suggestions and raises an important point (#7), which refers to the need to validate the drugs in a validated mouse sepsis model. Of course all other issues raised need to be convincingly addressed. Please do not hesitate to contact me in case you would like to discuss/clarify any of the points listed by the referees.

REVIEWER REPORTS

Reviewer #2:

The paper discusses a new method for inferring drug impact of diseases which integrates general gene expression data with curated lists of genes related to specific drugs and diseases. The method is applied to identify potential drugs for sepsis by exploiting an interesting observation: Sepsis seems

to impact adults much more than children. By analyzing expression differences for sepsis patients between childrens and adults the authors identify differentially regulated pathways in their network and use these to infer potential useful drugs for this condition. The identified drugs were validated using literature search and 10 of them were experimentally tested, of which 5 showed significant impact in animal models.

The main methodological advance in this paper is the use of an unbiased set of array data to identify relationship between gene signatures related to drugs and conditions. The use of such unbiased set is not completely new, and a number of large(er) GEO studies have been published. Still, the use of such approach for drug repurposing is likely novel. However, since this is the focus the authors need to do a better job at justifying why such approach makes sense and validating its general success. The current analysis for sepsis is interesting, but does not conclusively address this issue of utility. Thus, an important question is whether the findings presented represent a new direction for sepsis treatment

Specific comments:

1. The paper is very well written, discusses an important problem and contains some nice ideas. However, the overall direction the paper is advocating has been utilized for over a decade now. The main difference between prior approaches and the current one is the reliance on a larger, though non specific, expression set for inferring relationships between the various signatures collected by the authors. Once the network is constructed the selection of potential drugs is similar to prior methods. So the main question for this paper is: How much does the addition of such large expression compendium contribute to the ability to accurately identify relevant drugs. The authors try to address this indirectly, by comparing their method to variants of prior methods on the specific condition they analyzed, but since the paper advocates a new methodology based on the analysis they performed they need to better evaluate the global relevance of the network they derive. Can they comment on connections in the network that are unique? Surprising? Any way to validate some of these?
2. The method for constructing the network is also not completely well justified. Specifically, the authors require correlation (or corrected correlations) between the lists based on all 60K array expression datasets they use. Why is this justified? What about pathways that are correlated in some of the datasets but not in the others? These may score poorly globally, but may be very important for specific conditions (similar to the idea of bi-clustering, where conditions can also be divided). A global correlation score seem very restrictive.
3. Why did you use only 58K arrays? This seems to be based on a 2012 dataset collected as part of your previous 2013 paper. But since then GEO has grown dramatically. It would be useful to increase the datasets used to reflect the current size and number of arrays in GEO. Can you construct a PDN based on all current human GEO array / RNA-Seq datasets? How would the results based on that PDN compare to the current results you have?
4. How was the sepsis metadata curated (page 17)? Was this done using some automated process or manually? If the latter, how can this approach be generalized to other diseases? Would users need to manually search all expression datasets to identify experiments of interest?
5. Why weren't LINCS derived drug pathways included in the PDN? Based on the writeup, the only drugs the authors used were from CMAP itself since the pharma database did not provide up / down lists. But LINCS provides such lists for all drugs they tested. Why weren't these included in the PDN and used to select the most relevant drugs? The current network seems to be very restricted if it can only use one source of drug signature information.
6. The validation using curated data for the 45 top predicted drugs is a bit biased since it requires manual inspection. The authors are correct that no gold standard information exists. But what about using partial gold standard? For example, what is the overlap with known sepsis or sepsis related drugs in pharamagkb? Or any other database that list such drugs? This would be a much better initial validation if available.
7. Why did you perform the DE genes comparison (page 23) on a single array platform? Why not integrate across all platform you used (for example, those in the intersection of all of these platforms or based on some weighted version of their rank across the platforms)?
8. For the DE genes comparison (page 23), how did you select which nodes in the PDN they mapped to? How many PDN nodes were selected for each of the 400 or so genes you had?
9. Anyway to explain some of the findings / drugs you identified by the set of genes in your model? Since these were identified based on expression data, why not try and determine the molecular mechanisms that led you to this predictions in order to associate the predictions with actual genes?

Can these genes be related to the drugs you identified?

10. As you note, most of the drugs tested (and even a larger percentage of the ones that were validated) are known to be related to inflammatory response and treatment (all except for one of the validated drugs). Its true that not all inflammatory drugs are indeed effective, as you note, but still, why not select more interesting drugs for validation? The fact that several inflammation related drugs worked is nice, but its not clear if this fully utilizes the power of the proposed approach

Reviewer #3:

This is an interesting paper. The authors make use of publicly available data to investigate a well-known recurrent observation, namely that young children recover from a sepsis event, in the intensive care unit, much faster than adults and elderly. Based on their work they identify a number of key pathways explaining the observations, and they try to couple chemical compounds to these pathways, some of which indeed in a mouse model of endotoxemia (not sepsis) seem to hold therapeutic power. Here follow some major issues that have to be dealt with.

1. The authors suggest that sex hormones could explain part of the differences observed between children and adults, as mortality rates increase after puberty. Did the authors check for significant differences between male and female (adult) patients? Or did they perform such experiments in mice?
2. For the experimental validation of 10 drugs, 4 predicted drugs were not used as such, but were replaced by compounds with a similar chemical structure, as shown in Table S7. Do the authors have evidence that these replacements perform in the same (biological and pharmacological) way as the original, predicted compound. Can the authors explain why these 'similar' drugs were not predicted by their proposed method?
3. The authors refer to table S8 for the details of doses and administration routes of the tested drugs. However this table shows multiple concentrations for several drugs, and it is not clearly written in the paper what was now actually done experimentally. Are these concentration ranges, and if so, which concentrations were actually used to obtain the data shown in figure 3. Or do the authors mean that different concentrations were injected at the two stated time points (24h before and together with the LPS)?
4. The authors claim that a lethal dose of LPS was administered and give an 'example' of such a dose, but they do not specify how much LPS the mice received.
5. In the legend to figure 3, what do the authors mean with "Significant differences in concentration between pre- and post-pubertal mice are labeled with ..." while no such different mice are mentioned in the paper. Also is the result of the mantel-cox test then not shown on the figure?
6. In the Materials & methods subsection "Selection of pathways for analysis by PDN". The authors describe the 'N' metric, which was lower for the D cluster. Based in the information provided, does this mean that those pathways were upregulated in at least 70% of the samples in children? If so why is this value not 80% like for the other 3 clusters? If not, the 'N' should be explained in more detail.
7. The LPS endotoxemia model basically is a model of systemic inflammatory response syndrome (SIRS), not sepsis. To validate the drugs within the context of sepsis (what this whole study is about), the authors should test the compounds in a validated mouse sepsis model, for example the cecal ligation and puncture model, generally considered as the gold standard model for mouse sepsis work.

Minor issues

8. Please check the labels and readability of the Table, column heads etc.
9. The text, especially pathway names on Fig.1, are difficult to read due to pixilation
10. The quality of the parts in Fig. 4 varies (parts 2,3 & 5)

1st Revision - authors' response

2 March 2018

We would like to thank the reviewers for the helpful suggestions that led us to make the paper more cohesive and thorough. Here we address the reviewers' comments point-by-point. We were able to

address almost all concerns and believe that the paper is substantially improved. We hope you consider it acceptable for publication in MSB.

REVIEWER #2

The paper discusses a new method for inferring drug impact of diseases which integrates general gene expression data with curated lists of genes related to specific drugs and diseases. ... The main methodological advance in this paper is the use of an unbiased set of array data to identify relationship between gene signatures related to drugs and conditions. The use of such unbiased set is not completely new, and a number of large(er) GEO studies have been published. Still, the use of such approach for drug repurposing is likely novel. However, since this is the focus the authors need to do a better job at justifying why such approach makes sense and validating its general success. The current analysis for sepsis is interesting, but does not conclusively address this issue of utility. Thus, an important question is whether the findings presented represent a new direction for sepsis treatment

Authors Response:

We appreciate these thoughtful comments. We take a slightly different view of the main ‘methodological advance’ of the paper. Namely, we find that the powerful and well-cited CMap approach can be substantially improved by a pathway-centered elaboration on the original CMap concept. Our paper puts this to the test, both by curation and by experimentation. We are indeed encouraged by the positive findings in both of these tests. However, we must agree that whether the findings will truly represent ‘a new direction for sepsis treatment’ is unanswered.

Sepsis is enormously complex, and has a well-earned reputation as a graveyard for drug discovery attempts. One of many criticisms of past efforts is on over-reliance on flawed animal models (e.g. the mouse). We sought to reduce this negative by starting with human data. This choice, along with the novel pathway-based drug identification scheme, constitutes a unique contribution to the field. In our view, establishing the efficacy of promising leads would require two distinct and major efforts: 1) assessment of the drug leads in a better animal model of sepsis, e.g. the rabbit (for reasons detailed below, see answer to point 7, reviewer #3) and 2) subsequent clinical trials for any validated leads. Both of these are enormous tasks beyond the scope of this paper. Nevertheless, we consider that the information shared will be useful to the community, if only in its efforts to improve the ‘reversal of signature’ CMap-based approach by (1) developing signatures from human data as opposed to animal models; (2) incorporating the pathway approach; (3) using in silico/experimental benchmarking of leads; and (4) sharing these in detail when publishing.

As suggested by these general comments and the specific critiques from both reviewers (below) , we now include several new text sections to make these points more clearly and clarify “why such approach makes sense and validating its general success”. These edits are detailed in the replies to specific comments that follow.

Specific comments:

1. The paper is very well written, discusses an important problem and contains some nice ideas.the main question for this paper is: How much does the addition of such large expression compendium contribute to the ability to accurately identify relevant drugs. ...since the paper advocates a new methodology based on the analysis they performed they need to better evaluate the global relevance of the network they derive. Can they comment on connections in the network that are unique? Surprising? Any way to validate some of these?

Authors Response:

We have addressed the reviewer’s request for more information about the network by expanding the methods section to include greater characterization and discussion of the PDN. The first new subsection, “PDN network topology,” describes the overall node distribution of the network. We discuss how the degree distribution does not follow a power law and the PDN does not follow a scale-free topology. We reference a new supplementary figure (Fig EV2) showing the degree distribution and cumulative degree probability of the network. The second new subsection, “Biological interpretation of PDN pathway relationships,” provides a high level view of the major network components. We have clustered the network to define the major connected pathway

modules, described the most significant modules in the text, and have provided a supplementary table with a list of all of the modules (Table EV6). Although there are no outright ‘surprising’ relationships, it is clear that pathways that are closer to each other share biologically tractable and/or interesting relationships. We note that pathways that are clustered appear to have known or logical relationships. We provide examples in the narrative and note that Cluster 1 shows pathways with shared functionality in stress, immune response and cancers, concordant with the literature. We can now turn to the reviewer’s point regarding validation. Differences in pathway expression between the higher survival child samples vs. the lower survival adult samples include the somewhat surprising dominance of up-regulated metabolic pathways in children (e.g. steroid hormone biosynthesis, statin pathway activity, and cytochrome P450 activity). We agree with the reviewer’s interest in pursuing these intriguing findings more deeply. However, we must plead that the complexity of validation precludes including such efforts in this work. For example, the up-regulation of the statin pathway among children immediately suggests a potential benefit for statins in sepsis or severe infections. In fact, this has already been the focus of much controversial literature. Though animal model data and epidemiology point to better survival in statin-treated individuals (Lee et al, 2017; Takano et al, 2011, other studies in critically ill patients show these drugs to have no benefit {National Heart, 2014 #252). Advocates of statins as a treatment for sepsis point out that these negative studies in critically ill patients may be adding the intervention too late into the course of an ICU admission--when patients are already very ill and at high risk of death . These advocates call for further testing of early-intervention with statins, using appropriate animal models. We discuss the difficulties in animal models for sepsis and our position that future work should use a more human-like model (e.g. the rabbit) in response # 7 to reviewer 3 below. However, given that we don’t have any actual data of our own to offer, we would prefer not to expand on the more circumspect consideration of these issues already in the Discussion section.

NEW OR MODIFIED TEXT: MATERIALS AND METHODS

Page 24 (paragraph #3) – Page 26 (paragraph #1):

Network characterization

PDN network topology:

The PDN degree distribution and degree cumulative probability are shown in Fig EV2. At first glance, the degree distribution of the PDN plotted on a log-log scale may be considered roughly linear--indicative of a scale-free network following a power law distribution with gamma of approximately 0.61. However, the cumulative probability plot, which would also be linear on a log-log scale under scale-free conditions, clearly demonstrates significant divergence from a power law distribution. An exponential distribution or power law with exponential cutoff provides more reasonable but not exact fits. Scale-free networks were thought to be a common characteristic of biological networks (Albert, 2005), generally rationalized by the hypothesis that such networks are robust to random breaks and facilitate rapid inter-node communication by short average path lengths and high clustering coefficients. However, as higher resolution experimental data has become available, the general scale-free nature of biological networks has been increasingly questioned (Lima-Mendez & van Helden, 2009). The degree distribution of the PDN is evidence of a denser structure than would be expected from a power-law distributed network and provides an additional example of departure from scale-free topology. It should be noted that the presence of the drug perturbation and disease-state signatures in the PDN would also be expected to disrupt the structural characteristics of a network based on canonical pathways alone.

Biological interpretation of PDN pathway relationships.

We wished to establish whether pathways are correlated within the PDN in biologically meaningful ways. We created a sub-network consisting of all pathways from the network. Markov Clustering (van Dongen, 2000) of pathway-pathway correlation using the Graphia Pro environment (Kajeka.com) generated 38 biologically consistent clusters containing between 6 and 34 pathways as nodes (Table EV6). Pathway nodes vary in degree from 244 (Pathway. {PRKACA,33} (StaticModule)) to 4 (Pathway.TNF-alpha/NF-kB Signaling Pathway (Wikipathways)). Interpretation of cluster membership is complicated by the fact that only a partial understanding of known functional relationships between pathways exists. We have begun to address this challenge in a separate study of global pathway relationships (Pita-Juarez et al, PLoS Computational Biology, *in press*).

Clusters show pathways related by function. As cluster size decreases, functions become more specific. One example of this can be seen in our largest cluster, Cluster 1, which contains

pathways sharing functionality across cellular responses to stress, infections and cancers, B and T Cell receptor signaling pathways, as well as Tuberculosis, Leishmaniasis, and Toxoplasmosis pathways. Other clusters are enriched for pathways in lipid metabolism (Cluster 3), cell cycle and DNA replication (Cluster 6), immune signaling (Cluster 8), DNA repair and replication and RAS family genes (Cluster 10), extracellular matrix function (Cluster 12), and electron transport, respiratory chain function, oxidative phosphorylation and Parkinson's disease (Cluster 14). Functional relationships between pathways structured in this clustering approach may be insightful in terms of providing data driven context to known relationships. For instance, shared functionality in immune-mediated mechanisms of stress surveillance in cancer is an existing observation (Seelige et al, 2018).

2. *The method for constructing the network is also not completely well justified. Specifically, the authors require correlation (or corrected correlations) between the lists based on all 60K array expression datasets they use. Why is this justified? What about pathways that are correlated in some of the datasets but not in the others? These may score poorly globally, but may be very important for specific conditions (similar to the idea of bi-clustering, where conditions can also be divided). A global correlation score seem very restrictive.*

3. *Why did you use only 58K arrays? This seems to be based on a 2012 dataset collected as part of your previous 2013 paper. But since then GEO has grown dramatically. It would be useful to increase the datasets used to reflect the current size and number of arrays in GEO. Can you construct a PDN based on all current human GEO array / RNA-Seq datasets? How would the results based on that PDN compare to the current results you have?*

Authors Response to both points #2 and #3 here:

We agree with the general points made here that additional manipulations and variations could potentially provide further improvements and insights. Along with the methods mentioned by the reviewer in point 2, the good news is that there are multiple other creative approaches offered in the literature and often available as web-based tools. We offer 3 examples (Henriques et al, 2017; Li et al, 2017; Pathan et al, 2015) but the list could easily be expanded to include 10, or 50 or more, depending on how we interpret the reviewers comment.

Unfortunately, the time and resources required to explore these varied approaches are limited. We restricted our dataset to a single platform to ensure that the gene coverage and probe representation was the same across all experiments. We agree that the continued expansion of GEO and the new wave of RNA-seq data provide great opportunities. In the future, integrating expression data to build networks from cross-platform correlation will provide increased confidence in edges, especially when collating disease-specific background sets. We believe this report is a solid foundation for further improvement, and that the suggestions made here should be among those tested. Nevertheless, a deep dive into even a subset (if not all) of these methodologies is beyond the scope of an already data-rich paper. Similarly, we expect that re-construction of a PDN based on "all current human GEO array / RNA-Seq datasets" would likely lead to a set of drug leads that would at least be partially different, and would require a whole new round of experimentation to match the current data.

Ultimately, we believe that the priority for future endeavors should be on the general point raised by the reviewer in his/her initial comments: namely, testing our current list of drug leads *in vivo* (in a better animal model) and in clinical trials.

As suggested, we now include explicit mention of these ideas and suggestions in the paper for the benefit of the reader and to improve the perspective on potential future directions.

NEW OR MODIFIED TEXT: DISCUSSION

Page 16 (End of paragraph #2, continuing to paragraph #1 of Page 17):

It is also important to note that the quantity of available transcriptomic data (microarrays & RNA-Seq datasets) has grown (and continues to expand daily) since the PDN base network was

first constructed. A rich collection of other approaches to data-mining exists in the literature (Henriques et al, 2017; Li et al, 2017; Pathan et al, 2015). Integration of other analytic strategies might offer additional insights, and this exploration merits future consideration. Similarly, an expanded PDN based on a current version of the LINCS database (now accessed at clue.io) might provide additional power.

4. How was the sepsis metadata curated (page 17)? Was this done using some automated process or manually? If the latter, how can this approach be generalized to other diseases? Would users need to manually search all expression datasets to identify experiments of interest?

Authors Response:

The sepsis data were curated through a combination of automated steps (database searches for keywords) and manual work to compile and identify whether the required metadata were available (e.g. age of subject providing sample in a given dataset). In some cases, we needed to contact authors to obtain such information. The reviewer's comments emphasize well-documented limitations of meta-analysis or data-mining approaches: the need for extensive 'data-cleaning' and the maddening variability in how well public datasets are annotated by their submitting scientists. We do think this approach can be generalized to other diseases, but success will depend on many factors, including the availability of sufficient metadata to address the question of interest. This question is partially answered by past examples of similar successful data-mining approaches used for a variety of diseases (Duan et al, 2015; Toro-Dominguez et al, 2014; van den Akker et al, 2014).

To address the final question posed by the reviewer, there is no definitive answer as to whether users would need to manually search all expression datasets. We can imagine successful studies based on a subset of available datasets if the signal for the particular question is strong enough. However, we can also imagine failures if a sufficient number of datasets are not included (at the expense of laborious, but necessary, manual curation.)

As suggested, we now include explicit clarification and discussion of these points in the Materials and Methods section.

NEW OR MODIFIED TEXT: MATERIALS AND METHODS

Page 19 (paragraph #3):

This curation process involved both automated steps (e.g. database searches for keywords) as well as manual work to compile and identify whether the required metadata were available (e.g. age of subject providing sample in a given dataset). In some cases, the authors of the individual studies were contacted to obtain such information. The retrieved metadata were filtered, standardized and the relevant annotations of interest extracted (i.e. age, sepsis status, gender, data locations).

5. Why weren't LINCS derived drug pathways included in the PDN? Based on the writeup, the only drugs the authors used were from CMAP itself since the pharma database did not provide up / down lists. But LINCS provides such lists for all drugs they tested. Why weren't these included in the PDN and used to select the most relevant drugs? The current network seems to be very restricted if it can only use one source of drug signature information.

Authors Response:

Similar to our response to points #2 and #3, we agree with the value of this suggestion but consider it an endeavor better suited for a subsequent study. Specific limitations with the LINCS datasets include some lack of stability regarding what is available to the non-specialist user (i.e., the recent conversion from the LINCS website to the new clue.io was accompanied by a reduction in the data output from >20 cell lines to 9 core cell lines). We were already in the middle of our experimental validation attempts when this change was announced, so we proceeded with the data that was fully in hand and already in relatively wide use in the scientific community. The LINCS group is to be commended for constantly working to improve this resource, and the full data needed to do what the reviewer proposes is indeed now available at clue.io (subject to any future upgrades). Similar to the other good ideas from this reviewer, we will include the LINCS data in our future efforts, but we consider the data in this manuscript, obtained using the CMAP list, to be a proof-of-principle and

thus a valuable finding in its own right. This position is now explicitly stated in the discussion (see new text in copy of Discussion section provided in the answer to reviewer points #2 & #3 above)

6. The validation using curated data for the 45 top predicted drugs is a bit biased since it requires manual inspection. The authors are correct that no gold standard information exists. But what about using partial gold standard? For example, what is the overlap with known sepsis or sepsis related drugs in pharamagkb? Or any other database that list such drugs? This would be a much better initial validation if available.

Authors Response:

We agree and wish there were such drugs to perform validation studies. Unfortunately, there are no approved host-directed drugs for sepsis. Therapy presently relies solely on antimicrobials for the infection, as well as hemodynamic and respiratory support. The failure of past drug discovery efforts for sepsis is a major motivation for our work.

7. Why did you perform the DE genes comparison (page 23) on a single array platform? Why not integrate across all platform you used (for example, those in the intersection of all of these platforms or based on some weighted version of their rank across the platforms)?

Authors Response:

By using one array platform, we aimed to reduce one of the many confounders known to be limitations when performing DE gene comparisons across different studies. We once again must agree with the reviewer that multiple other comparisons are possible. We also note that almost all of the pediatric samples available were done on this single array platform (GPL570 aka U133 Plus 2.0- which has a robust high degree of gene coverage), so including a variety of arrays for the adult samples might have introduced imbalanced confounding to only one of the groups being analyzed. Ultimately, each meta-analysis method comes with its own ‘baggage’ and so we chose to use this simple version in the proof-of-principle spirit behind this comparison.

8. For the DE genes comparison (page 23), how did you select which nodes in the PDN they mapped to? How many PDN nodes were selected for each of the 400 or so genes you had?

Authors Response:

The DE genes were split into up- and down-regulated gene signatures. Each of these gene signatures was incorporated into the PDN as a new network node. The edges created between these new nodes and the CMap drugs were then used to prioritize drugs. The reviewer is correct to point out that this was insufficiently described in the materials and methods section. In response, the text has been expanded to include a subsection explicitly titled “Determining CMap drugs associated with an individual gene signature.”

NEW OR MODIFIED TEXT: MATERIALS AND METHODS

Page 28 (paragraph #2, continuing until the end of paragraph #1 on Page 29):

Determining CMap drugs associated with an individual gene signature

To compare PDN functionality based on a network analysis of a target cluster of pathways with standard single gene-list based analysis, we queried the PDN directly with a differential signature derived from adult and child transcriptomes (see above). The top 500 up- and down-regulated probes from a comparison of children vs. adults using datasets limited to a single array platform (GPL570) were matched to 427 and 405 up- and down-regulated genes in CMap. These gene signatures were incorporated into the PDN and ranked for positive or negative association with CMap drug signatures by a similar approach as the Pathprint cluster pathways described above. The differentially expressed genes were split into up-regulated and down-regulated gene-sets. These gene-sets were then introduced into the PDN as two new nodes and evaluated separately. The p-values of the edges connecting each of the new nodes to the separate CMap up- and down-regulated gene signatures were aggregated for each drug. This calculation helped to quantify whether the up- and down-regulated components of the adult vs. child differentially expressed signatures were positively or negatively associated with each CMap drug. Then, an

overall positive or negative correlation between a differential gene signature and a CMap drug was determined by combining the p-values calculated for the up-regulated and down-regulated parts of the signature. The top 45 negatively associated drugs were selected for further validation.

9. Any way to explain some of the findings / drugs you identified by the set of genes in your model? Since these were identified based on expression data, why not try and determine the molecular mechanisms that led you to this predictions in order to associate the predictions with actual genes? Can these genes be related to the drugs you identified?

Authors Response:

We are not sure exactly how to address this question.

The predictions we made were based on pathways identified using the Pathprint method, based on genes expressed by the children or adult groups. Perhaps the reviewer is suggesting that we take the gene list defining a pathway that is up-regulated in adults and then analyze whether this gene list informs the mechanism(s) of action of the predicted/validated drugs identified for possible inhibition/reversal of this adult phenotype. This would presume the existence of a set of genes that define the mechanism(s) of action of the lead drug. There are gene expression profiles available for the drugs in CMap, but these vary substantially between the cell lines and time points studied. The PDN analysis that led to identification of the drug identifies connections between multiple pathways and a given drug, so it is not clear how to go forward on this particular analytical path.

Alternatively, perhaps the reviewer has in mind an analysis of the top active drugs to see which pathway connections they share in the PDN. If this is the case, we can report that we did pursue this line of inquiry. To build a network of pathways down-regulated by the top drugs showing *in vivo* benefit, we used the following criteria: 1) include pathways that were negatively associated with each of the top drugs (i.e. they were negatively correlated with the CMap up-regulated signature and positively correlated with the CMap down-regulated signature); 2) include pathways that were negatively associated with at least two of the drugs. In this line of inquiry, we found a number of groupings of interest (e.g., pathways negatively associated with chlorpromazine as well as camptothecin--as well as other combinations) and one ‘failure to group’ which is also interesting: amitriptyline did not associate with any of the other drugs. Review of the scores of these pathways in the original Pathprint analysis showed that the paired drugs associated with pathways that are up-regulated or unregulated in adults and should be moving the pathways toward the desirable down-regulated status seen in children. In essence, we found these results to be a reassuring, but somewhat circular confirmation of the method that do not merit inclusion in this paper. The finding of a single outlier of sorts, amitriptyline, does hint that different mechanisms, or perhaps unique signaling reflected in transcriptome profiles, may characterize this drug. These are interesting beginnings that are being actively pursued in more depth. We hope these comments address the question the reviewer had in mind here.

10. As you note, most of the drugs tested (and even a larger percentage of the ones that were validated) are known to be related to inflammatory response and treatment (all except for one of the validated drugs). Its true that not all inflammatory drugs are indeed effective, as you note, but still, why not select more interesting drugs for validation? The fact that several inflammation related drugs worked is nice, but its not clear if this fully utilizes the power of the proposed approach

Authors Response:

We agree that there are other drug leads of potential interest with different mechanisms of action that should be tested. However, because these are further down in rank, we would need to either: 1) have expanded the pool of drugs to be tested (and hence the number of experimental trials), or 2) ‘cherry-picked’ other drugs of interest. The latter would reasonably be criticized as biased if successful or a mistaken waste of effort if not. In both cases we would need to return to rank-based testing, either at reviewer’s instruction for the former scenario of success or at our own choice for the latter failure scenario.

We take heart from the reviewer's comment that this (and other suggestions made earlier) may more "fully utilize(s) the power of the proposed approach." We expect our future efforts to do just that.

REVIEWER #3

This is an interesting paper. The authors make use of publicly available data to investigate a well-known recurrent observation, namely that young children recover from a sepsis event, in the intensive care unit, much faster than adults and elderly. Based on their work they identify a number of key pathways explaining the observations, and they try to couple chemical compounds to these pathways, some of which indeed in a mouse model of endotoxemia (not sepsis) seem to hold therapeutic power. Here follow some major issues that have to be dealt with.

1. The authors suggest that sex hormones could explain part of the differences observed between children and adults, as mortality rates increase after puberty. Did the authors check for significant differences between male and female (adult) patients? Or did they perform such experiments in mice?

Authors Response:

This is an interesting point indeed. However, there were a number of reasons that we did not try to compare transcriptome differences between male and female adult patients. First, the epidemiologic data supporting greater survival of children vs. adults is based on data using both sexes. Since our goal was to use this age-based difference to identify drug leads, we focused on comparisons of the two age groups, and not the sexes of the subjects within those groups. The analysis of sex-based differences in sepsis in adults is further complicated by known differences in overall rate and in comorbidities that might skew analyses by sex. To comprehensively investigate this question, we expect that an even larger number of patients with well-annotated samples would be necessary.

As for mice, we have indeed investigated sex differences, albeit in a different model that has some relevance to the issues here. One of the main epidemiologic findings of relative survival advantages for children from severe infectious disease comes from the 1918 influenza pandemic. We have modeled this in mice and reported the findings in a recent publication (Suber & Kobzik, 2017). Both male and female pre-pubertal mice survive flu that is highly fatal to postpubertal mice of both sexes. The pubertal survival difference seen in this mouse model appears to depend on the effects of low-level elevations of estrogen during puberty in both sexes. Similar findings apply in female mice studied in an endotoxemia model, i.e. pre-pubertal mice survive better and pubertal estrogen mediates greater susceptibility to death among post-pubertal mice (Joachim et al, 2017). However, the latter study did not include male mice.

As suggested by this point, we now include a brief mention of some of these ideas for the interested reader in the Discussion, as excerpted below:

NEW OR MODIFIED TEXT: DISCUSSION

Page 13 (End of paragraph #1, continuing on to the top of Page 14):

The change in resistance is linked to the pubertal transition, suggesting a key role for sex hormones. Indeed other experimental studies from our laboratory support this idea (Joachim et al, 2017; Suber & Kobzik, 2017), and indicate that this topic merits further investigation in human studies.

2. For the experimental validation of 10 drugs, 4 predicted drugs were not used as such, but were replaced by compounds with a similar chemical structure, as shown in Table S7. Do the authors have evidence that these replacements perform in the same (biological and pharmacological) way as the original, predicted compound. Can the authors explain why these 'similar' drugs were not predicted by their proposed method?

Authors Response:

We do not have direct experimental evidence that they perform the same way. However, the drugs used are very similar to the original predicted compounds: e.g. vinpocetine is a synthetic derivative

of the vinca alkaloid vincamine, topotecan is a water-soluble analogue of camptothecin, etc. The specific reasons for the use of similar compounds are now expanded upon in the Materials and Methods and mentioned briefly in the legend for Table EV5 (formerly Table S7.)

For three of the four substitutions (topotecan, chlorpromazine, amitriptyline), our rationale was driven by publications showing survival benefit in animal models using the similar substitute drug-- data that did not exist for the original predicted compounds. For the final pair, vincamine and vinpocetine, there are no published data demonstrating a survival benefit for either drug. However, there are data showing that vinpocetine has some anti-inflammatory activity (Jeon et al, 2010). No similar data was found for vincamine, which motivated our choice to use vinpocetine.

The final question is insightful, but we do not have a satisfactory answer. We can speculate and invoke cell-line or drug-specific differences in pharmacokinetics or gene expression responses, but a more succinct response is that we don't know. The absence of published data similar to ours (where ranked lists of drug candidates from CMap or elsewhere are curated *in silico* and then tested experimentally) precludes us from finding this same phenomenon in other studies, which might at least inform how often it happens. We hope this paper will stimulate other investigators to produce and share such data.

As suggested by this point, we now include clarification and discussion of these issues in the Materials and Methods and Table EV5 (formerly Table S7) as excerpted below:

NEW OR MODIFIED TEXT: MATERIALS AND METHODS

Page 30 (paragraph #2):

Substitutions with highly similar compounds were made for some of the predicted drugs. For three of the four substitutions made (topotecan, chlorpromazine, amitriptyline), the rationale was driven by the existence of publications showing survival benefit in animal models using the substituted drug ((Brand et al, 2008; Rialdi et al, 2016; Villa et al, 1995), respectively). These data did not exist for the original predicted drugs. For the final pair, vincamine/vinpocetine, there are no published data demonstrating a survival benefit, but there are data showing that vinpocetine has some anti-inflammatory activity (Jeon et al, 2010). No similar data was found for vincamine, thus motivating our choice of vinpocetine.

NEW OR MODIFIED TEXT: TABLE EV5

READ ME_TableEV2.txt

Substitutions with highly similar compounds were made for some of the predicted drugs. The rationale is detailed in the Materials and Methods.

3. The authors refer to table S8 for the details of doses and administration routes of the tested drugs. However this table shows multiple concentrations for several drugs, and it is not clearly written in the paper what was now actually done experimentally. Are these concentration ranges, and if so, which concentrations were actually used to obtain the data shown in figure 3. Or do the authors mean that different concentrations were injected at the two stated time points (24h before and together with the LPS)?

Authors Response:

As suggested, we have clarified these points by additional text in Table EV7 (formerly table S8).

NEW OR MODIFIED TEXT: TABLE EV7

READ ME_TableEV3.txt

Mice were treated with the predicted drug candidates for 24 hours before and on the day of LPS administration using the specific delivery routes, doses, and vehicle compositions detailed above. For some compounds, two doses were tested and the similar results combined to create the summary graphs used in Fig 3 (topiramate and noscapine, 25 and 100 mg/kg; ethacrynic acid 10 and 50 mg/kg; each, N=4 animals per dose).

4. *The authors claim that a lethal dose of LPS was administered and give an 'example' of such a dose, but they do not specify how much LPS the mice received.*

Authors Response:

As suggested, we now state explicitly how much LPS the mice received in both the Materials and Methods, as well as the Fig 3 legend.

NEW OR MODIFIED TEXT: MATERIALS AND METHODS

Page 30 (paragraph #3):

...C57bl/6 female mice (5 weeks old, Charles River, Wilmington, DE) were injected with a high lethality dose (e.g. 38-40 µg/g) of *E. coli* LPS (L3755; Lot: 123M4096V) (Sigma-Aldrich, St. Louis, MO) between hours 5 and 7 of the light period in the animal facility (12-2PM).

NEW OR MODIFIED TEXT: FIGURE 3 LEGEND

Page 42:

Fig 3. ...C57bl/6 female mice were injected with a high-lethality dose of *E. coli* LPS (38-40 µg/g) followed by a subcutaneous injection of sterile saline.

5. In the legend to figure 3, what do the authors mean with "Significant differences in concentration between pre- and post-pubertal mice are labeled with ..." while no such different mice are mentioned in the paper. Also is the result of the mantel-cox test then not shown on the figure?

Authors Response:

Thank you for spotting this typo. We have removed this erroneous text and replaced it as excerpted below:

NEW OR MODIFIED TEXT: FIGURE 3 LEGEND

Page 42:

Fig 3. Significant differences in concentration between drug and vehicle-treated pre- and post-pubertal mice are labeled with **** (p<0.0001), *** (p<0.001), ** (p<0.01), or * (p<0.05). Percent survival was compared using a log rank Mantel Cox test.

6. In the Materials & methods subsection "Selection of pathways for analysis by PDN". The authors describe the 'N' metric, which was lower for the D cluster. Based in the information provided, does this mean that those pathways were upregulated in at least 70% of the samples in children? If so why is this value not 80% like for the other 3 clusters? If not, the 'N' should be explained in more detail.

Authors Response:

The reviewer is absolutely correct. We lowered the stringency for cluster D to increase the yield of pathways to be similar to those for the other 3 clusters.

As suggested, this is clarified now by the text excerpted below:

NEW OR MODIFIED TEXT: MATERIALS AND METHODS

Page 21 (paragraph #2):

For cluster D, use of the 80% criterion produced only one pathway for evaluation. Hence, the threshold was lowered to allow inclusion of the four pathways that were up-regulated in at least 70% of the samples from children and un-regulated in adults.

7. The LPS endotoxemia model basically is a model of systemic inflammatory response syndrome (SIRS), not sepsis. To validate the drugs within the context of sepsis (what this whole study is about), the authors should test the compounds in a validated mouse sepsis model, for example the cecal ligation and puncture model, generally considered as the gold standard model for mouse sepsis work.

Authors Response:

This is the most challenging point to address! We agree with the reviewer's advocacy of using additional sepsis models to further support our findings. There are a number of reasons why we have not done this and why we take the position that such additional experimentation is beyond the scope of this paper.

First, all murine sepsis models are generally considered to be imperfect—each having both positive and negative aspects. For example, they elicit different transcriptomic responses and often do not include the supportive care regimens typical of human patients (e.g. fluid resuscitation, antibiotics, oxygen therapy, etc.). Mice are much more resistant to LPS than humans (~5-6 logs more LPS/kg needed to cause mortality in mice). Moreover, even the “gold standard” sepsis model, cecal ligation and puncture (CLP), is recognised as being technically difficult and variable; different responses are elicited from lab-to-lab or even from person-to-person within a given lab. Second, the (admittedly also imperfect) LPS model used in these studies, does fulfill an important criterion: it mirrors the pre- vs. post-pubertal human epidemiology that interests us, as detailed in our recent publication (Joachim et al, 2017). Thus, we believe that the endotoxemia model is a sufficient tool to begin to investigate the underlying mechanisms driving pre-pubertal resistance. Any results in another mouse model would also be subject to these reasonable criticisms, and hence would be of limited value. Ultimately, we would like to expand pre-pubertal resistance model to a different species that is more similar to humans in sensitivity to endotoxin and sepsis—the rabbit. We especially note that a similar resistance to mortality from LPS in pre-pubertal vs. pubertal rabbits has been reported (Watson & Kim, 1963) although this finding was not the focus of the cited study. While we endorse the reviewer's suggestion, expanding our studies into other species and other models of sepsis will take considerable effort and time beyond what is appropriate for this study.

As suggested by this comment, we have expanded our discussion of this point to better address this issue and this text is excerpted below:

NEW OR MODIFIED TEXT: DISCUSSION**Page 11 (End of paragraph #1, paragraph #2 and continuing on to the top of Page 12):**

Moreover, even the “gold standard” sepsis model, cecal ligation and puncture (CLP), is recognized as being technically difficult and variable; different responses are elicited from lab-to-lab or even from person-to-person within a given lab. The (admittedly also imperfect) LPS model used in these studies, does fulfill an important criterion: it mirrors the pre- vs. post-pubertal human epidemiology that interests us, as detailed in our recent publication (Joachim et al, 2017). Thus, we believe that the endotoxemia model is a sufficient tool to begin our investigation of the underlying mechanisms driving pre-pubertal resistance. Ultimately, we would like to expand the pre-pubertal resistance model to a species that is more similar to humans in sensitivity to endotoxin and sepsis—the rabbit. We especially note that a similar resistance to mortality from LPS in pre-pubertal vs. pubertal rabbits has been reported (Watson & Kim, 1963) although this finding was not the focus of the cited study.

8. Please check the labels and readability of the Table, column heads etc.Authors Response:

We checked all table labels and edited some to improve readability.

9. The text, especially pathway names on Fig.1, are difficult to read due to pixilation.Authors Response:

We are puzzled by this comment. For us, the figure and text for pathway names in Fig 1 are quite legible and not pixelated on our screens and when printed out using a standard ink-jet printer. We will be glad to work with the editorial office to improve the figure if deemed necessary. We note the other reviewer did not raise this concern, and that we were able to see the problem pointed out in Fig 4 in the next point. We will supply such figures in PDF format and ensure better clarity.

10. The quality of the parts in Fig 4 varies (parts 2,3 & 5).

Authors Response:

As suggested, the overall quality of Fig 4 was improved.

References

- Albert R (2005) Scale-free networks in cell biology. *J Cell Sci* **118**: 4947-4957
- Brand V, Koka S, Lang C, Jendrossek V, Huber SM, Gulbins E, Lang F (2008) Influence of Amitriptyline on Eryptosis, Parasitemia and Survival of *Plasmodium Berghei*-Infected Mice. *Cell Physiol Biochem* **22**: 405-412
- Duan J, Sanders AR, Moy W, Drigalenko EI, Brown EC, Freda J, Leites C, Goring HH, Mgs, Gejman PV (2015) Transcriptome outlier analysis implicates schizophrenia susceptibility genes and enriches putatively functional rare genetic variants. *Hum Mol Genet* **24**: 4674-4685
- Henriques R, Ferreira FL, Madeira SC (2017) BicPAMS: software for biological data analysis with pattern-based biclustering. *BMC Bioinformatics* **18**: 82
- Jeon KI, Xu X, Aizawa T, Lim JH, Jono H, Kwon DS, Abe J, Berk BC, Li JD, Yan C (2010) Vinpocetine inhibits NF-kappaB-dependent inflammation via an IKK-dependent but PDE-independent mechanism. *Proc Natl Acad Sci U S A* **107**: 9795-9800
- Joachim R, Suber F, Kobzik L (2017) Characterising Pre-pubertal Resistance to Death from Endotoxemia. *Sci Rep* **7**: 16541
- Lee CC, Lee MG, Hsu TC, Porta L, Chang SS, Yo CH, Tsai KC, Lee M (2017) A Population-Based Cohort Study on the Drug-Specific Effect of Statins on Sepsis Outcome. *Chest*
- Li Y, Jourdain AA, Calvo SE, Liu JS, Mootha VK (2017) CLIC, a tool for expanding biological pathways based on co-expression across thousands of datasets. *PLoS Comput Biol* **13**: e1005653
- Lima-Mendez G, van Helden J (2009) The powerful law of the power law and other myths in network biology. *Mol Biosyst* **5**: 1482-1493
- Pathan M, Keerthikumar S, Ang CS, Gangoda L, Quek CY, Williamson NA, Mouradov D, Sieber OM, Simpson RJ, Salim A, Bacic A, Hill AF, Stroud DA, Ryan MT, Agbinya JI, Mariadason JM, Burgess AW, Mathivanan S (2015) FunRich: An open access standalone functional enrichment and interaction network analysis tool. *Proteomics* **15**: 2597-2601
- Rialdi A, Campisi L, Zhao N, Lagda AC, Pietzsch C, Ho JSY, Martinez-Gil L, Fenouil R, Chen X, Edwards M, Metreveli G, Jordan S, Peralta Z, Munoz-Fontela C, Bouvier N, Merad M, Jin J, Weirauch M, Heinz S, Benner C et al (2016) Topoisomerase 1 inhibition suppresses inflammatory genes and protects from death by inflammation. *Science* **352**: aad7993
- Seelige R, Searles S, Bui JD (2018) Mechanisms regulating immune surveillance of cellular stress in cancer. *Cell Mol Life Sci* **75**: 225-240
- Suber F, Kobzik L (2017) Childhood tolerance of severe influenza: a mortality analysis in mice. *Am J Physiol Lung Cell Mol Physiol* **313**: L1087-L1095
- Takano K, Yamamoto S, Tomita K, Takashina M, Yokoo H, Matsuda N, Takano Y, Hattori Y (2011) Successful treatment of acute lung injury with pitavastatin in septic mice: potential role of glucocorticoid receptor expression in alveolar macrophages. *J Pharmacol Exp Ther* **336**: 381-390
- Toro-Dominguez D, Carmona-Saez P, Alarcon-Riquelme ME (2014) Shared signatures between rheumatoid arthritis, systemic lupus erythematosus and Sjogren's syndrome uncovered through gene expression meta-analysis. *Arthritis Res Ther* **16**: 489
- van den Akker EB, Passtoors WM, Jansen R, van Zwet EW, Goeman JJ, Hulsman M, Emilsson V, Perola M, Willemsen G, Penninx BW, Heijmans BT, Maier AB, Boomsma DI, Kok JN, Slagboom PE, Reinders MJ, Beekman M (2014) Meta-analysis on blood transcriptomic studies identifies consistently coexpressed protein-protein interaction modules as robust markers of human aging. *Aging Cell* **13**: 216-225

van Dongen S (2000) A Cluster Algorithm for Graphs. Retrieved from <http://micansorg/mcl/lit/INS-R0010psZ>: 1-42

Villa P, Sartor G, Angelini M, Sironi M, Conni M, Gnocchi P, Isetta AM, Grau G, Buurman W, van Tits LJ (1995) Pattern of cytokines and pharmacomodulation in sepsis induced by cecal ligation and puncture compared with that induced by endotoxin. *Clin Diagn Lab Immunol* **2**: 549-553

Watson DW, Kim YB (1963) Modification of Host Responses to Bacterial Endotoxins. I. Specificity of Pyrogenic Tolerance and the Role of Hypersensitivity in Pyrogenicity, Lethality, and Skin Reactivity. *J Exp Med* **118**: 425-446

2nd Editorial Decision

17 April 2018

Thank you again for submitting your work to Molecular Systems Biology. Unfortunately, despite a series of reminders we did not manage to obtain a report from reviewer #2. We have however heard back from reviewer #3 who is now satisfied with the modifications made and thinks that the study is suitable for publication.

Before we formally accept your manuscript for publication, we would ask you to address the following remaining editorial issues.

REVIEWER REPORT

Reviewer #3:

This is a highly original approach to address the complex problem of sepsis. It results in novel insights and drug targets. The authors have also replied, with satisfaction, to most of my previous comments. The inclusion of the CLP mouse sepsis model in their story would have been an added value, but the authors have chosen not to follow this suggestion and I can live with their argumentation.

Corresponding Author Name: Lester Kobzik
Journal Submitted to: Molecular Systems Biology
Manuscript Number: MSB-17-7998R